# A PX-BAR protein Mvp1/SNX8 and a dynamin-like GTPase Vps1 drive endosomal recycling

**Sho W Suzuki, Akihiko Oishi, Nadia Nikulin, Jeff R Jorgensen, Matthew G Baile, Scott D Emr***

Weill Institute for Cell and Molecular Biology and Department of Molecular Biology and Genetics, Cornell University, Ithaca, United States

**Abstract** Membrane protein recycling systems are essential for maintenance of the endosome-lysosome system. In yeast, retromer and Snx4 coat complexes are recruited to the endosomal surface, where they recognize cargos. They sort cargo and deform the membrane into recycling tubules that bud from the endosome and target to the Golgi. Here, we reveal that the SNX-BAR protein, Mvp1, mediates an endosomal recycling pathway that is mechanistically distinct from the retromer and Snx4 pathways. Mvp1 deforms the endosomal membrane and sorts cargos containing a specific sorting motif into a membrane tubule. Subsequently, Mvp1 recruits the dynamin-like GTPase Vps1 to catalyze membrane scission and release of the recycling tubule. Similarly, SNX8, the human homolog of Mvp1, which has been also implicated in Alzheimer's disease, mediates formation of an endosomal recycling tubule. Thus, we present evidence for a novel endosomal retrieval pathway that is conserved from yeast to humans.

**\*For correspondence:**
sde26@cornell.edu

**Competing interest:** The authors declare that no competing interests exist.

## Introduction

Recycling sorting components (eg, receptors, SNAREs, transporters, etc) in the endo-lysosome system are essential for the normal assembly and function of the lysosome. The best-characterized recycling cargo is yeast Vps10. Vps10, the first member of the sortilin receptor family, is a transmembrane (TM) protein receptor that sorts carboxypeptidase Y (CPY) into vesicles at the Golgi (*Marcusson et al., 1994*). After CPY-containing vesicles are transported to the endosome, the endosome matures and fuses with the vacuole, delivering soluble CPY to the vacuole lumen. Unlike CPY, which is released from the Vps10 receptor in the endosome, Vps10 is not delivered to the vacuole. It is recycled from the endosome back to the Golgi by retromer, making Vps10 available for additional rounds of CPY sorting. Retromer is an evolutionarily conserved protein coat complex composed of five proteins: Vps5, Vps17, Vps26, Vps29, and Vps35 (*Figure 1A–B*, *Figure 1—figure supplement 1A*; *Seaman et al., 1997*; *Seaman et al., 1998*; *Suzuki et al., 2019*). It deforms the endosomal membrane to form cargo-containing recycling tubules/vesicles. In humans, the loss of retromer function alters the cellular localization of hundreds of TM proteins. A mutation in *VPS35* has been associated with Alzheimer's disease (AD) and Parkinson's disease (*Vilariño-Güell et al., 2011*; *Zimprich et al., 2011*; *Rovelet-Lecrux et al., 2015*; *Small and Petsko, 2015*). Retromer also has been identified as an essential host factor for severe acute respiratory syndrome coronavirus 2 (SARS-nCoV2) infection (*Daniloski et al., 2021*). Thus, retromer-mediated endosomal recycling has been linked to diverse pathologies (*Teasdale and Collins, 2012*; *Small and Petsko, 2015*; *McMillan et al., 2017*).

Sorting nexin (SNX) is an evolutionarily conserved protein family comprising a Phox homology (PX) domain that binds to phosphatidylinositol 3-phosphate (PI3P), allowing its selective interaction with endosome (*Burda et al., 2002*; *Carlton et al., 2004*). Some SNXs also possess a Bin-amphiphysin-Rvs167

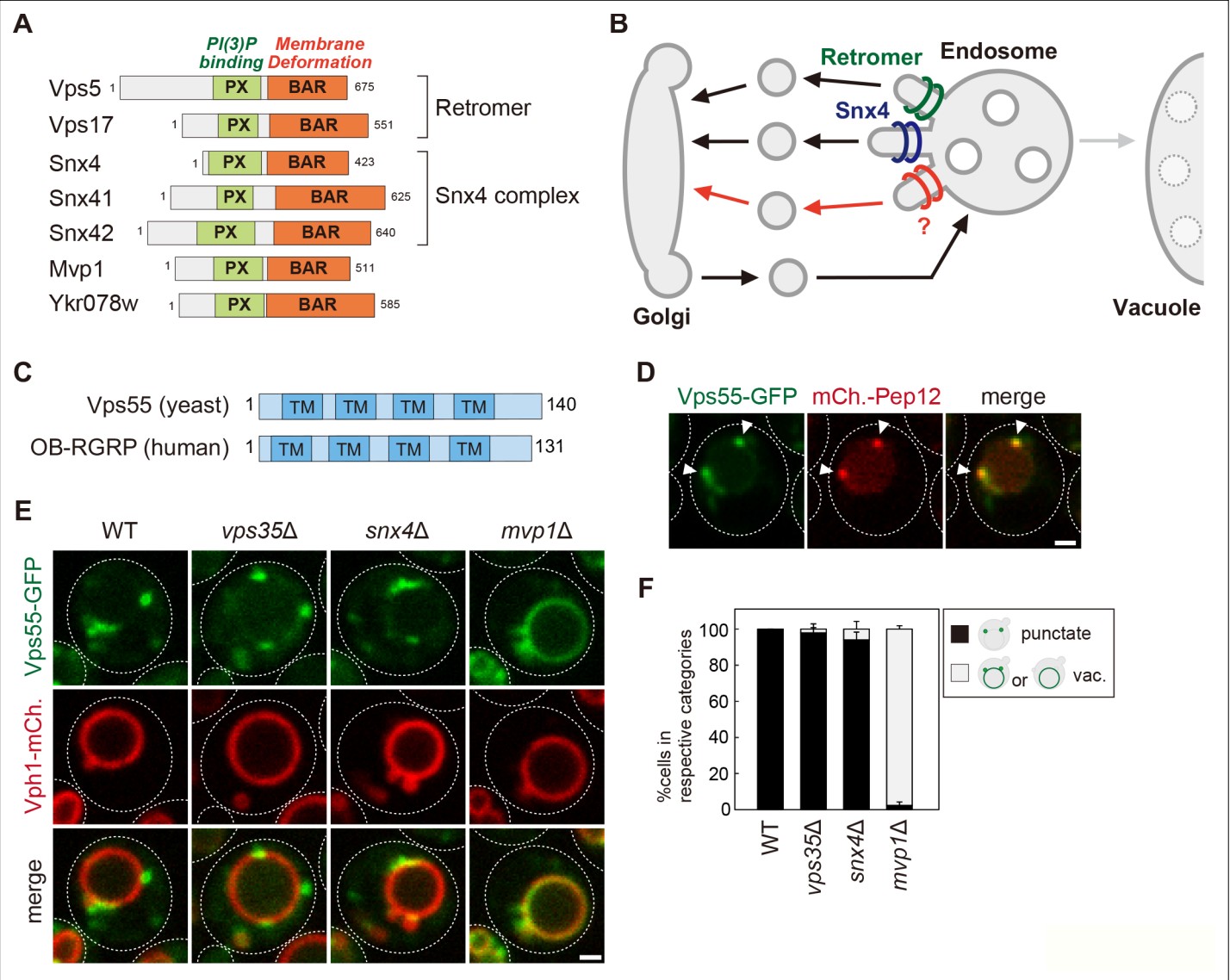

**Figure 1.** The endosomal localization of Vps55 requires Mvp1. (**A**) Schematic of SNX-BAR proteins in yeast. (**B**) Model of endosomal recycling pathways in yeast. (**C**) Schematic of Vps55 and OB-RGRP. (**D**) Vps55-GFP localization. The mCherry-Pep12 serves as an endosomal marker. (**E**) Vps55-GFP localization in wild-type (WT), *vps35Δ* (retromer mutant), *snx4Δ* (Snx4 complex mutant), and *mvp1Δ*. (**F**) Quantification of Vps55-GFP localization from three independent experiments. N = >30 cells. Scale bar: 1 μm.

The online version of this article includes the following source data and figure supplement(s) for figure 1:

**Source data 1.** Source data associated with *Figure 1F*.

**Source data 2.** The list of localization altered in retromer mutants. The list of endosomal transmembrane proteins whose localization was examined in vps35Δ cells.

**Figure supplement 1.** The localization of endosomal membrane proteins.

**Figure supplement 1—source data 1.** Source data associated with *Figure 1—figure supplement 1D*.

(BAR) domain that induces/stabilizes membrane curvature. These SNXs have been classified as the SNX-BAR subfamily (*Cullen, 2008*). Previous studies have identified seven yeast and twelve mammalian SNX-BAR proteins (*Figure 1A*). In budding yeast, Vps5 (SNX1/2 in humans), Vps17 (SNX5/6/32 in humans), Snx4 (SNX4 in humans), Snx41 (SNX7 in humans), Snx42 (SNX30 in humans), Mvp1 (Snx8 in humans), and Ykr078w belong to this family. Vps5 and Vps17 function in membrane binding/deformation as retromer subunits (*Horazdovsky et al., 1997*; *Seaman et al., 1997*; *Seaman et al., 1998*). Snx4 forms a complex with either Snx41 or Snx42 and mediates retromer-independent membrane

protein recycling from the endosome and the vacuole (*Figure 1B*, *Figure 1—figure supplement 1A*; *Hettema et al., 2003*; *Suzuki and S.D, 2018*).

Mvp1 is a yeast SNX-BAR protein identified as a multicopy suppressor of dominant-negative *VPS1* mutations (*Ekena and Stevens, 1995*). A previous study has proposed that Mvp1 is required for retromer-mediated recycling (*Chi et al., 2014*). Interestingly, two single-nucleotide polymorphisms within the SNX8 (the human homolog of MVP1) gene locus are associated with late-onset AD (*Rosenthal et al., 2012*). *Xie et al., 2019* report that the SNX8 expression level is significantly lower in AD patients and APP/PS1 AD mouse brain. Overexpression of SNX8 suppresses the accumulation of fragments of amyloid precursor protein (Aβ). Additionally, patients lacking SNX8 were shown to have heart development defects, intellectual disability, learning and language delay, and severe behavioral problems related to the hyperactive-impulsive and inattentive area (*Vanzo et al., 2014*; *Rendu et al., 2014*; *Mastromoro et al., 2020*). Although SNX8 has been linked to several diseases, its molecular function is still poorly characterized. Here, we show that Mvp1 recycles membrane proteins in a retromer- and Snx4-independent manner in yeast. We also demonstrate that SNX8 mediates formation of endosomal recycling tubules in humans. Thus, we propose Mvp1 mediates a conserved endosomal recycling pathway that is mechanistically distinct from the retromer and Snx4 pathways. This study reveals that yeast has three major SNX-BAR endosomal recycling pathways: retromer, Snx4, and Mvp1.

## Results

### The endosomal localization of Vps55 requires Mvp1

To characterize the function of Mvp1 and test the hypothesis that Mvp1 is involved in membrane protein recycling, we tested the requirement of Mvp1 function for endosomal TM cargo proteins. To maintain endosomal localization, these proteins need to be recycled back to the Golgi mainly by the retromer before the endosome fuses with the vacuole. Indeed, the endosomal t-SNARE Pep12 accumulates on the vacuole membrane in retromer-defective *vps35Δ* cells, because retromer-mediated endosome-to-Golgi retrograde trafficking is impaired (*Figure 1—figure supplement 1B*). We hypothesized that TM proteins that remain properly localized to the endosome in a retromer-independent manner might be cargo for Mvp1. We examined several endosomal TM proteins' localization in *vps35Δ* cells (*Figure 1—source data 2*) and found Vps55 still localized on the endosome even in retromer mutants.

Vps55 is an endosomal membrane protein involved in the fusion of the endosome with the vacuole (*Figure 1C*; *Belgareh-Touzé et al., 2002*; *Schluter et al., 2008*). Vps55 is evolutionarily conserved and its human homolog is OB-RGRP (OB-R gene-related protein), which is responsible for downregulation of specific leptin receptors (*Couturier et al., 2007*). When we expressed chromosomally tagged Vps55-GFP, it was localized on punctate structures that colocalized with the endosomal membrane protein mCherry-Pep12, but not with the trans-Golgi protein Sec7-mCherry, as previously reported (*Figure 1D*, *Figure 1—figure supplement 1C, D*; *Belgareh-Touzé et al., 2002*; *Schluter et al., 2008*). In retromer mutants (*vps35Δ*), it was still localized on the endosome (*Figure 1E and F*). The Snx4 complex is known to mediate retromer-independent endosomal recycling (*Hettema et al., 2003*), but Vps55-GFP was also localized on the endosome in Snx4 complex mutants (*snx4Δ*, *snx41Δ*, and *snx42Δ*) (*Figure 1E and F*, *Figure 1—figure supplement 1E*). Strikingly, Vps55-GFP accumulated on the vacuole membrane in *mvp1Δ* cells (*Figure 1E and F*). Vps55-GFP localized on the endosome in cells lacking Ykr078w, another poorly characterized SNX-BAR protein (*Figure 1—figure supplement 1E*). These results suggest that the endosomal localization of Vps55 requires Mvp1.

### Mvp1 is an endosomal coat complex for Vps55 recycling

Mvp1 has been characterized as a retromer-associated SNX-BAR protein (*Chi et al., 2014*). However, its function in the retromer-independent pathway has never been studied. Endogenously expressed Mvp1-GFP localized to the endosome as previously reported (*Figure 2A*; *Chi et al., 2014*). To test if Mvp1 also localizes to the vacuole membrane, we examined the localization of Mvp1 in cells lacking Pep12, which is essential for endosome assembly (*Becherer et al., 1996*). In *pep12Δ* cells, Mvp1-GFP lost its endosomal localization and was distributed to the cytoplasm (*Figure 2—figure supplement 1A*), suggesting that Mvp1 specifically localizes on the endosome.

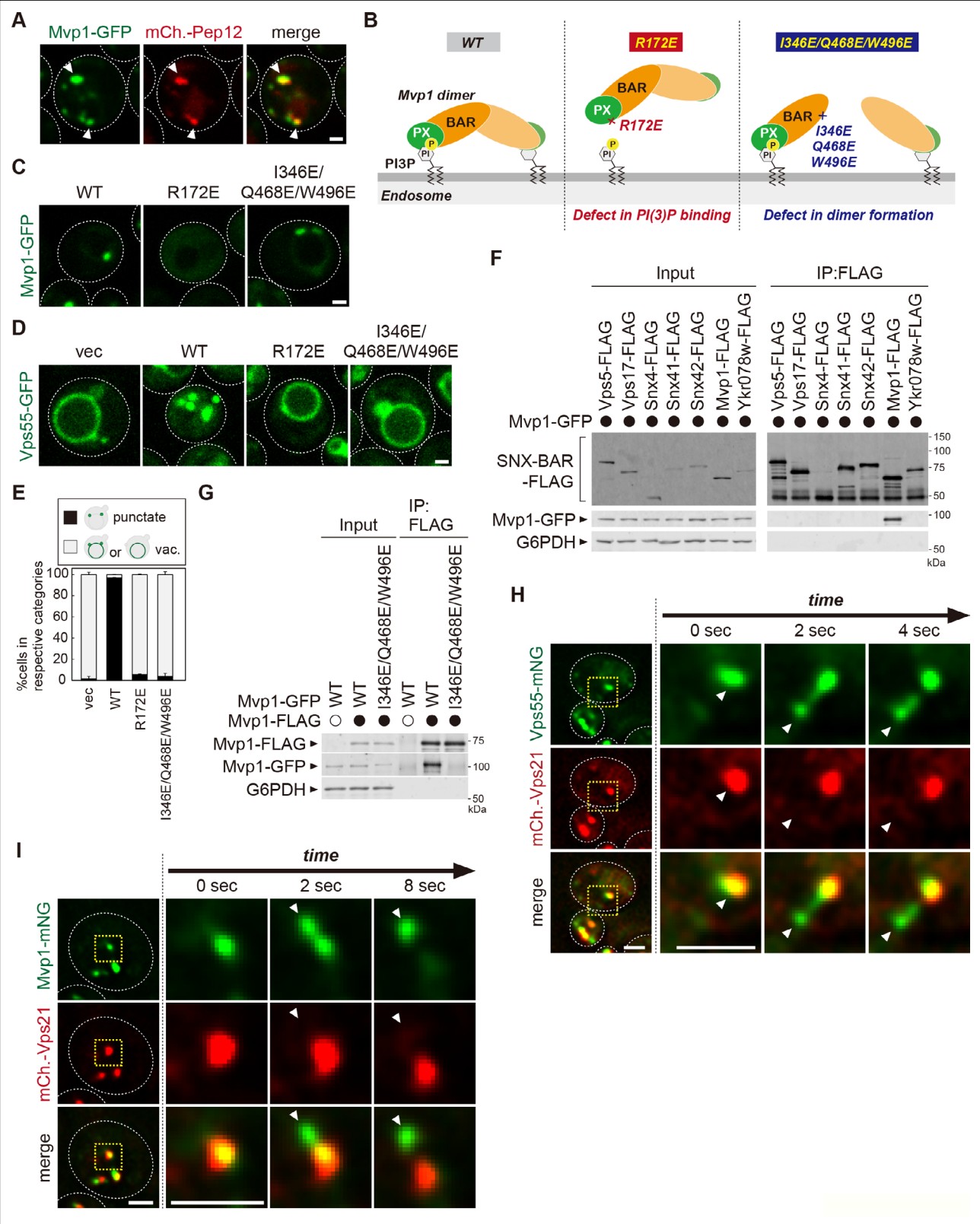

**Figure 2.** Mvp1 is an endosomal coat complex for Vps55 recycling. (**A**) Mvp1-GFP localization. The mCherry-Pep12 serves as an endosomal marker. (**B**) Schematic of Mvp1 mutants. (**C**) The localization of Mvp1-GFP mutants. (**D**) The localization of Vps55-GFP mutants. (**E**) Quantitation of Vps55-GFP localization of *mvp1* mutants from three independent experiments. N = >30 cells. (**F**) The binding of SNX-BAR proteins with Mvp1. FLAG-tagged SNX-BAR proteins were immunoprecipitated (IP) from cells expressing Mvp1-GFP, and the IP products were analyzed by immunoblotting using antibodies

*Figure 2 continued on next page*

*Figure 2 continued*

against FLAG, green fluorescent protein (GFP), and glucose-6-phosphate dehydrogenase (G6PDH). (**G**) The dimer formation of *mvp1* mutants. Mvp1-FLAG was immunoprecipitated from cells expressing Mvp1-GFP mutants, and the IP products were analyzed by immunoblotting using antibodies against FLAG, GFP, and G6PDH. (**H**) Live-cell imaging analysis of Vps55-mNeonGreen and mCherry-Vps21. (**I**) Live-cell imaging analysis of Mvp1-mNeonGreen and mCherry-Vps21. Scale bar: 1 µm.

The online version of this article includes the following video and figure supplement(s) for figure 2:

**Source data 1.** Source data associated with *Figure 2E*.

**Source data 2.** Uncropped gel images of *Figure 2F*.

**Source data 3.** Uncropped gel images of *Figure 2G*.

**Figure supplement 1.** The analysis of Mvp1.

**Figure supplement 1—source data 1.** Source data associated with *Figure 2—figure supplement 1D*.

**Figure 2—video 1.** Vps55-mNeonGreen decorated tubule emerged and detached from the endosome.

https://elifesciences.org/articles/69883/figures#fig2video1

The PX domain of Mvp1 binds to PI3P, allowing its specific localization on the endosome (*Figure 2B*). The conserved residue R172 located in the PX domain is responsible for PI3P binding (*Figure 2—figure supplement 1B, C*; *Chi et al., 2014*). The R172E mutation severely impaired the endosomal localization of Mvp1 (*Figure 2C*). When the R172E mutant was expressed in *mvp1Δ* cells, Vps55-GFP localized to the vacuole membrane (*Figure 2D and E*), indicating that the endosomal localization of Mvp1 is required for appropriate Vps55 localization.

In addition to the PX domain, Mvp1 has a BAR domain, which induces/stabilizes membrane curvature (*Figure 2B*). Since the BAR domain's membrane remodeling activity requires homo- or heterodimer formation (*Frost et al., 2009*), we tested which SNX-BAR protein makes a dimer with Mvp1. We expressed FLAG-tagged SNX-BAR proteins in yeast and examined their binding with Mvp1-GFP. Mvp1-GFP coimmunoprecipitated with Mvp1-FLAG, but not with other SNX-BAR proteins (*Figure 2F*), suggesting that Mvp1 forms a homodimer but not heterodimers. Based on the recent cryo-electron microscopy (cryo-EM) structure of the Mvp1 (*Sun et al., 2020*), we mutated conserved residues located on the dimer interface predicted to disrupt dimer formation (*Figure 2—figure supplement 1B*). The combination mutant I346E/Q468E/W496E had a defect in dimer formation and failed to rescue the mislocalization of Vps55-GFP in *mvp1Δ* cells (*Figure 2D–E and G*). We confirmed that I346E/Q468E/W496E mutants still localize on the endosome (*Figure 2C*). These results suggest that the membrane deformation activity of Mvp1 is required for the endosomal localization of Vps55.

To characterize how Mvp1 recycles Vps55, we performed live-cell imaging of Vps55-mNeonGreen with an endosomal Rab GTPase mCherry-Vps21 (*Horazdovsky et al., 1994*). The Vps55-mNeonGreen decorated tubules that emerged and detached from both Vps55-mNeonGreen- and mCherry-Vps21-positive endosomes (*Figure 2H*, *Figure 2—figure supplement 1D*). Similarly, we observed that Mvp1-mNeonGreen-positive but mCherry-Vps21-negative tubules also budded from the endosome (*Figure 2I* and *Figure 2—video 1*), which was consistent with a previous report (*Chi et al., 2014*). Based on these observations, we propose that Mvp1 is an endosomal coat complex for Vps55 retrieval.

## Mvp1 recognizes Vps55 through a specific sorting motif

To determine the sequence motif required for Vps55 retrieval, we performed a mutational analysis of the cytoplasmic region of Vps55. We generated a series of Vps55-GFP mutants in which three or four consecutive amino acids of the cytoplasmic region were replaced with alanine residues and examined their localization (*Figure 3A*, *Figure 3—figure supplement 1A*). Since 2–4 A and 5–8 A mutants failed to express, we were not able to analyze residues 2–8 of Vps55. Of the tested mutants, 60–63 A and 64–67 A mutants exhibited a severe defect in Vps55 recycling. In contrast, 68–71 A, 72–75 A, 133–136 A, and 137–140 A did not show a striking defect. Next, we mutated single residues to alanine in the 60–67 region (*Figure 3B*, *Figure 3—figure supplement 1B*). Of the mutants tested, Y61A, T63A, F66A, and M67A stabilized Vps55-GFP on the vacuole membrane. K60A and D65A mutants also showed a defect, but it could be due to the higher expression of Vps55-GFP. Then, we generated Y61A/T63A, F66A/M67A, and Y61A/T63A/F66A/M67A mutants. While Y61A/T63A and F66A/M67A mutants showed a partial defect in Vps55-GFP recycling, the Y61A/T63A/F66A/M67A mutant exhibited a severe defect (*Figure 3C and D*). To test whether the 61-YHTSDFM-67 region is

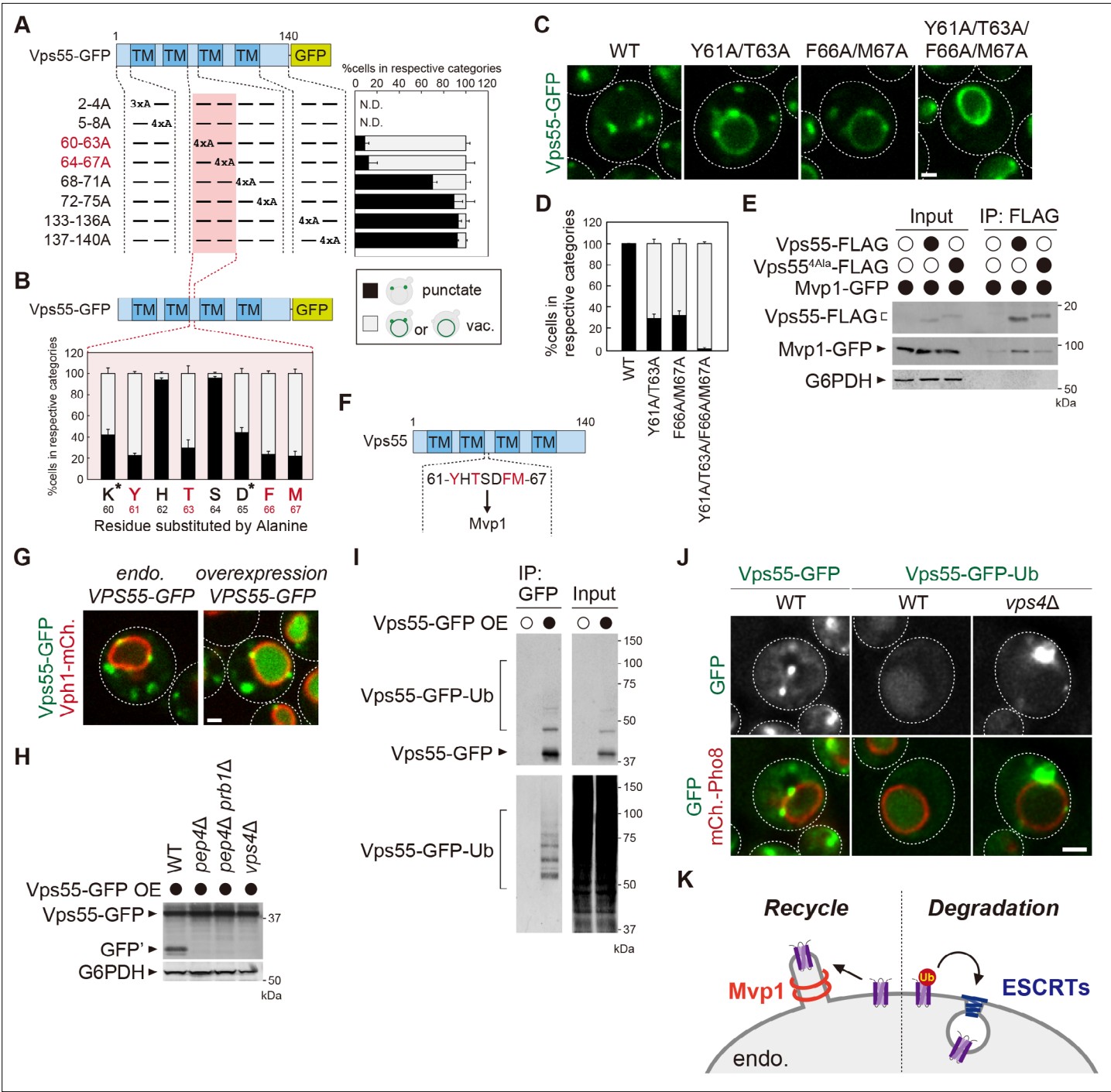

**Figure 3.** Mvp1 recognizes Vps55 through a specific sorting motif. (**A, B**, and **D**) Schematic of Vps55 mutational analysis and quantitation of Vps55-GFP mutant localization, from *Figure 3—figure supplement 1A* (**A**), *Figure 3—figure supplement 1B* (**B**), and *Figure 3—figure supplement 1C* (**D**). (**C**) The localization of Vps55-GFP mutants. (**E**) The Mvp1-Vps55 interaction in Vps55-FLAG mutants. Vps55-FLAG mutants were immunoprecipitated (IP) from cells expressing Mvp1-GFP, and the IP products were analyzed by immunoblotting using antibodies against FLAG, green fluorescent protein (GFP), and glucose-6-phosphate dehydrogenase (G6PDH). (**F**) Schematic of Vps55 and the residues facilitating its interaction with Mvp1. (**G**) The localization of overexpressed Vps55-GFP. (**H**) Vps55-GFP sorting in vacuolar hydrolases (*pep4Δ* and *pep4Δ prb1Δ*) and ESCRT (*vps4Δ*) mutants. Cell lysates expressing Vps55-GFP were analyzed by immunoblotting using antibodies against GFP and G6PDH. (**I**) The ubiquitination of overexpressed Vps55-GFP. Overexpressed Vps55-GFP was immunoprecipitated from yeast cells under denaturing conditions, and the IP products were analyzed by immunoblotting using antibodies against GFP and ubiquitin. (**J**) Vps55-GFP-Ub localization in ESCRT mutants. (**K**) Model of Vps55 recycling and degradation at the endosome. For all quantification shown in this figure, n = >30 cells from three independent experiments. Scale bar: 1 μm.

*Figure 3 continued on next page*

*Figure 3 continued*

The online version of this article includes the following figure supplement(s) for figure 3:

**Source data 1.** Source data associated with *Figure 3A*.

**Source data 2.** Source data associated with *Figure 3B*.

**Source data 3.** Source data associated with *Figure 3D*.

**Source data 4.** Uncropped gel images of *Figure 3E*.

**Source data 5.** Uncropped gel images of *Figure 3H*.

**Source data 6.** Uncropped gel images of *Figure 3I*.

**Figure supplement 1.** The analysis of Vps55.

**Figure supplement 1—source data 1.** Uncropped gel images of *Figure 3—figure supplement 1C*.

**Figure supplement 1—source data 2.** Uncropped gel images of *Figure 3—figure supplement 1D*.

**Figure supplement 1—source data 3.** Source data associated with *Figure 3—figure supplement 1F*.

**Figure supplement 1—source data 4.** Uncropped gel images of *Figure 3—figure supplement 1G*.

required for binding with Mvp1, we performed coimmunoprecipitation experiments. Mvp1-GFP was coprecipitated with the wild-type (WT) Vps55-FLAG, whereas the binding of the Y61A/T63A/F66A/M67A mutant was impaired (*Figure 3E*). Since the Y61A/T63A/F66A/M67A mutant localized on the vacuole membrane, we also examined the Mvp1-Vps55 interaction in cells lacking Vam3, which is an essential SNARE for the endosome to fuse with the vacuole. Mvp1-GFP was still coprecipitated with Vps55-FLAG, but not with Vps55$^{4Ala}$-FLAG (*Figure 3—figure supplement 1C*). These results suggest that 61-YHTSDFM-67 in Vps55 is required for Mvp1 recognition, although it is also possible that the Vps55-Mvp1 interaction is bridged by another protein (*Figure 3F*).

## Excess Vps55 is ubiquitinated and degraded rather than recycled

Many proteins are assembled into the protein complex of defined stoichiometry to function appropriately (*Yanagitani et al., 2017*; *Mena et al., 2018*). Some unassembled soluble proteins are ubiquitinated and then degraded through the proteasome. However, the fate of unassembled membrane proteins has not been well studied. Vps55 forms a complex with Vps68. This complex formation is required for the stability of both Vps55 and Vps68 (*Schluter et al., 2008*). When we overexpressed Vps55-GFP, it was sorted into the vacuole lumen and degraded (*Figure 3G*, *Figure 3—figure supplement 1D*). Consistent with this, Vps55-GFP processing was observed in WT cells by immunoblotting, whereas it was not in cells lacking vacuolar hydrolases *PEP4* and *PRB1* (*Figure 3H*). Overexpression of Vps68 suppressed degradation of Vps55-GFP (*Figure 3—figure supplement 1E, F*), suggesting that the unassembled excess pool of Vps55-GFP is degraded. Vacuolar degradative protein sorting is mainly mediated through the endosomal sorting complex required for transport (ESCRT) complex (*Henne et al., 2011*). Indeed, Vps55-GFP was not degraded in ESCRT-defective *vps4Δ* cells, suggesting that the ESCRT pathway is required for the vacuolar sorting of Vps55-GFP (*Figure 3H*). Cargo ubiquitination is a prerequisite for recognition by the ESCRT machinery (*Katzmann et al., 2001*). To ask whether Vps55 is ubiquitinated, we immunoprecipitated (IP) Vps55-GFP and were able to detect ubiquitinated forms of Vps55-GFP (*Figure 3I*). To ask if ubiquitination is sufficient for Vps55 degradation, we fused ubiquitin to Vps55-GFP. Vps55-GFP-Ub was sorted into the vacuole lumen in an ESCRT-dependent manner, even when expressed at endogenous levels (*Figure 3J*, *Figure 3—figure supplement 1G*). Collectively, we propose that excess Vps55 is ubiquitinated and degraded through the ESCRT pathway rather than recycled (*Figure 3K*).

## Mvp1 recruits dynamin-like GTPase Vps1 to catalyze membrane scission

Dynamin is a GTPase that has diverse roles in membrane remodeling events (*Ferguson and Camilli, 2012*; *Daumke et al., 2014*). Its function is best characterized in the context of clathrin-mediated endocytosis. During endocytic vesicle formation, a subset of BAR domain-containing proteins (FCHO1, FBP17, SNX9, endophilin, amphiphysin, etc), together with clathrin and its adaptor proteins, are recruited to the plasma membrane (PM). Subsequently, they induce membrane curvature to mediate the formation of a budding vesicle. Then, dynamin is recruited to its neck to mediate fission, thereby

releasing the vesicles (*Ferguson and Camilli, 2012*). To ask if dynamin catalyzes membrane scission in Mvp1-mediated recycling, we examined Vps55 localization in dynamin mutants. The yeast genome encodes four dynamin-related proteins, Dnm1, Vps1, Sey1, and Mgm1 (*Ford and Chappie, 2019*). Dnm1 and Vps1 have membrane scission activity, whereas Sey1 and Mgm1 do not. Dnm1 assembles on mitochondria to mediate mitochondrial fission (*Bleazard et al., 1999*; *Sesaki and Jensen, 1999*). Vps1 has been implicated in vacuole fusion/fission, endocytosis, peroxisome division, CPY sorting, and endosomal recycling (*Figure 4A*; *Peter et al., 2004*; *Kuravi et al., 2006*; *Rooij et al., 2010*; *Rooij et al., 2012*; *Hayden et al., 2013*; *Chi et al., 2014*). However, how Vps1 functions in these processes has not been characterized well. When Vps55-mNeonGreen was expressed in *vps1Δ* cells, it localized to the vacuole membrane, suggesting Vps1 is required for Vps55 recycling (*Figure 4B and C*).

To study the localization of Vps1, we tagged chromosomal *VPS1* with green fluorescent protein (GFP) using the LAP (70-residue localization and affinity purification) linker (*Guizetti et al., 2011*), because direct fusion of GFP to the C-terminus of *VPS1* interferes with function (*Chi et al., 2014*). Cells lacking Vps1 showed a growth defect at 37 °C, but Vps1-GFP-expressing cells grew as well as the WT cells, suggesting that this Vps1-GFP is functional (*Figure 4—figure supplement 1A*). Consistent with a previous report (*Varlakhanova et al., 2018*), Vps1-GFP colocalized with endosomal Nhx1-mCherry, but not with Golgi-localized Sec7-mCherry (*Figure 4—figure supplement 1B, C*, C). We also confirmed that Vps1-GFP colocalized with Mvp1-mRFP (*Figure 4—figure supplement 1D*). Several Vps1-GFP puncta did not colocalize with Mvp1-mRFP, presumably because it also localized on other organelles (ie, PM, peroxisome, etc). To study its dynamics, we performed live-cell imaging analysis of Vps1-GFP with mCherry-Vps21. The Vps1-GFP punctate structures on the mCherry-Vps21-marked endosomes were elongated and then divided (*Figure 4D* and *Figure 4—video 1*).

Upon dynamin assembly, guanosine-5'-triphosphate (GTP) hydrolysis induces conformational changes in the dynamin ring, leading to membrane fission (*Figure 4E*; *Ferguson and Camilli, 2012*). Subsequently, the dynamin ring is disassembled and reused for another round of scission. The K44A mutation in human Dynamin-1 impairs the GTP hydrolysis activity (*van der Bliek et al., 1993*; *Damke et al., 1994*). In this mutant, membrane scission of clathrin-coated vesicles is blocked. This lysine residue is widely conserved in dynamin-related GTPases, including Vps1 (*Figure 4F*, *Figure 4—figure supplement 1E*; *Varlakhanova et al., 2018*; *Tornabene et al., 2020*). To test if the GTP hydrolysis activity of Vps1 is required for Vps55 recycling, we examined Vps55 localization in *vps1^K42A* mutants. In this mutant, Vps55-mNeonGreen localized to the vacuole membrane (*Figure 4G and H*), suggesting that the membrane scission activity of Vps1 is required for Mvp1-mediated recycling. Next, we examined the localization of the Vps1^K42A mutant. Vps1^K42A-BFP showed punctate structures on the mCherry-Vps21-marked endosome, which colocalized with Mvp1-mNeonGreen (*Figure 4I*). These results suggest that the GTP hydrolysis activity of Vps1 is required for Vps55 retrieval.

To test if Mvp1 is required for endosomal localization of Vps1, we examined its localization in *mvp1Δ* cells, but Vps1-GFP is still localized on endosomes (*Figure 4—figure supplement 1F*). Since Vps1 is required for retromer- and Snx4-mediated recycling, we expressed Vps1-GFP in *vps35Δ snx4Δ mvp1Δ* triple mutants, in which the retromer, Snx4, and Mvp1 complexes are defective (*Chi et al., 2014*; *Lukehart et al., 2013*). Since the vacuole morphology was extremely fragmented in *vps35Δ snx4Δ mvp1Δ* cells (*Figure 4—figure supplement 1G*), it was difficult to determine cargo localization in these mutants. However, we found that the vacuole fragmentation phenotype was partially rescued by supplementing choline or ethanolamine, which are used for lipid synthesis. Under these conditions, we found that Vps1-GFP lost its punctate pattern and instead localized to the cytoplasm in *vps35Δ snx4Δ mvp1Δ* cells (*Figure 4J and K*). We also confirmed that Vps1-GFP maintained its endosomal localization in *vps35Δ* cells and *snx4Δ* cells (*Figure 4K*, *Figure 4—figure supplement 1F*). Since SNX-BAR proteins require PI3P for their recruitment to the endosomal membrane, we examined Vps1-GFP localization in *vps34Δ* cells. It lost its punctate localization and shifted to the cytoplasm (*Figure 4—figure supplement 1H*, I). On the other hand, consistent with a previous report (*Ekena and Stevens, 1995*), we observed that Mvp1-mNeonGreen was still localized on the endosome even in *vps1Δ* cells (*Figure 4—figure supplement 1J*). We introduced a mutation in the PI3P-binding site of Mvp1 (R172E), which altered its punctate localization in *vps1Δ* cells, confirming the endosomal localization of Mvp1 (*Figure 4—figure supplement 1K*, L). These results suggest that the SNX-BAR proteins, including Mvp1, are required for the endosomal localization of Vps1.

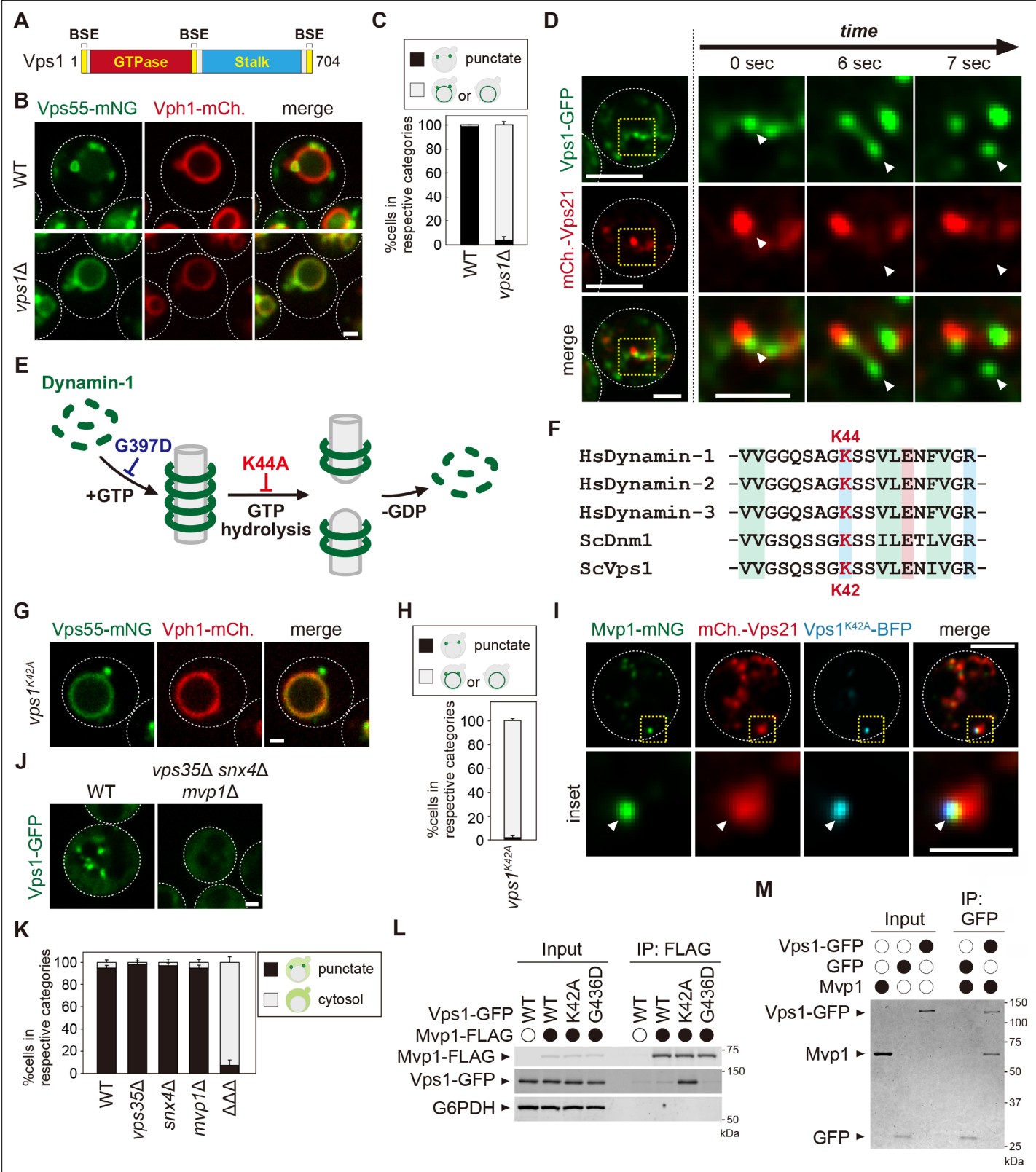

**Figure 4.** Mvp1 recruits dynamin-like GTPase Vps1 to catalyze membrane scission. (**A**) Schematic of Vps1. (**B**) Vps55-mNeonGreen localization in WT and *vps1Δ* cells. (**C**) Quantitation of Vps55-mNeonGreen localization, from **B**. (**D**) The live-cell imaging of Vps1-GFP. The mCherry-Vps21 serves as an endosomal marker. (**E**) Model of Dynamin-1-mediated membrane fission. (**F**) Sequence comparison of residues required for guanosine-5'-triphosphate (GTP) hydrolysis in *Homo sapiens* Dynamin-1, *H. sapiens* Dynamin-2, *H. sapiens* Dynamin-3, *Saccharomyces cerevisiae* Dnm1, and *S. cerevisiae* Vps1.

*Figure 4 continued on next page*

*Figure 4 continued*

(**G**) Vps55-mNeonGreen localization in *vps1^K42A* mutants. (**H**) Quantitation of Vps55-mNeonGreen localization, from **G**. (**I**) The localization of Mvp1-mNeonGreen, mCherry-Vps21, and Vps1^K42A-BFP. (**J**) Vps1-GFP localization in wild-type (WT) and *vps35Δ snx4Δ mvp1Δ* triple mutants. (**K**) Quantitation of Vps1-GFP localization, from **J** and *Figure 4—figure supplement 1F*. (**L**) The Mvp1-Vps1 interaction. Mvp1-FLAG was immunoprecipitated (IP) from cells expressing Vps1-GFP mutants, and the IP products were analyzed using antibodies against FLAG, green fluorescent protein (GFP), and glucose-6-phosphate dehydrogenase (G6PDH). (**M**) In vitro binding assay between Mvp1 and Vps1-GFP. The proteins bound to anti-GFP magnetic beads were detected by Coomassie staining. For all quantifications shown in this figure, n = >30 cells from three independent experiments. Scale bar: 1 μm.

The online version of this article includes the following video and figure supplement(s) for figure 4:

**Source data 1.** Source data associated with *Figure 4C*.

**Source data 2.** Source data associated with *Figure 4H*.

**Source data 3.** Source data associated with *Figure 4K*.

**Source data 4.** Uncropped gel images of *Figure 4L*.

**Source data 5.** Uncropped gel images of *Figure 4M*.

**Figure supplement 1.** The analysis of Vps1.

**Figure supplement 1—source data 1.** Source data associated with *Figure 4—figure supplement 1I*.

**Figure supplement 1—source data 2.** Source data associated with *Figure 4—figure supplement 1L*.

**Figure 4—video 1.** The Vps1-GFP punctate structures on the endosome were elongated and then divided.
https://elifesciences.org/articles/69883/figures#fig4video1

To ask if the oligomerization of Mvp1 is required for Vps1 recruitment to the endosome, we examined Vps1-GFP localization in *mvp1^R346E/Q468E/W496E* mutants, which have a defect in its oligomerization (*Figure 2B*). When we expressed Mvp1 in *vps35Δ snx4Δ mvp1Δ* cells, Vps1-GFP exhibited its endosomal localization. However, it lost its punctate localization in *vps35Δ snx4Δ mvp1Δ* cells expressing Mvp1^R346E/Q468E/W496E mutants (*Figure 4—figure supplement 1M*). These results suggest that Mvp1 oligomerization is required for the recruitment of Vps1.

Next, we analyzed the binding of Vps1 mutants to Mvp1. We found that the Vps1^K42A mutant interacts with Mvp1 more strongly than WT Vps1 (*Figure 4L*). The G436D mutant has a defect in the assembly of Vps1 (*Figure 4E*, *Figure 4—figure supplement 1E*, N), but the binding of this mutant was barely detected. We also prepared recombinant Mvp1 and Vps1-GFP and subjected them to in vitro binding assays. Mvp1 coprecipitated with Vps1-GFP, but not with GFP, suggesting that Mvp1 directly binds to Vps1 (*Figure 4M*). Based on these results, we propose that Mvp1 recruits the dynamin-like GTPase Vps1 to the site of vesicle tubule formation to catalyze membrane scission.

## Mvp1 mainly mediates a retromer-independent endosomal recycling pathway

Mvp1 has been proposed to function together with the retromer complex, but no interaction between the retromer and Mvp1 has been observed (*Chi et al., 2014*). To ask if Mvp1 is associated with the retromer complex, we immunoprecipitated the retromer complex from yeast cell lysates. When we immunoprecipitated Vps5-FLAG, the other retromer subunits Vps17-HA, Vps26-Myc, Vps29, and Vps35 were coimmunoprecipitated, but Mvp1-GFP was not (*Figure 5—figure supplement 1A*). Conversely, when we immunoprecipitated Mvp1-FLAG, retromer subunits Vps5, Vps17-HA, Vps26, Vps29, and Vps35 were not coimmunoprecipitated (*Figure 5—figure supplement 1B*). Since *mvp1Δ* cells only exhibit a mild CPY-sorting defect (*Ekena and Stevens, 1995*), we re-evaluated retromer cargo localization in *mvp1Δ* cells. We examined the localization of Vps10 (CPY receptor on the Golgi and endosome), Kex2 (serine protease on the Golgi), and Neo1 (P4 type ATPase on the endosome) (*Figure 5A and B*, *Figure 5—figure supplement 1C-F*; *Marcusson et al., 1994*; *Voos and Stevens, 1998*; *Dalton et al., 2017*). Vps10-GFP, Kex2-GFP, and GFP-Neo1 accumulated on the vacuole membrane in *vps35Δ* cells, whereas they localized on punctate structures in *mvp1Δ* cells, as well as WT cells. These observations suggest that Mvp1 is not required for retromer-mediated recycling.

Next, we examined if Mvp1 and retromer form distinct recycling vesicles from the endosome (*Figure 5—figure supplement 1G*). To test this, we biochemically immunoisolated Vps55-containing vesicles and asked if Vps10, a retromer cargo, is also present. In this experiment, we tried to accumulate Vps55-containing recycling vesicles using a temperature-sensitive mutant of Sec18, which is

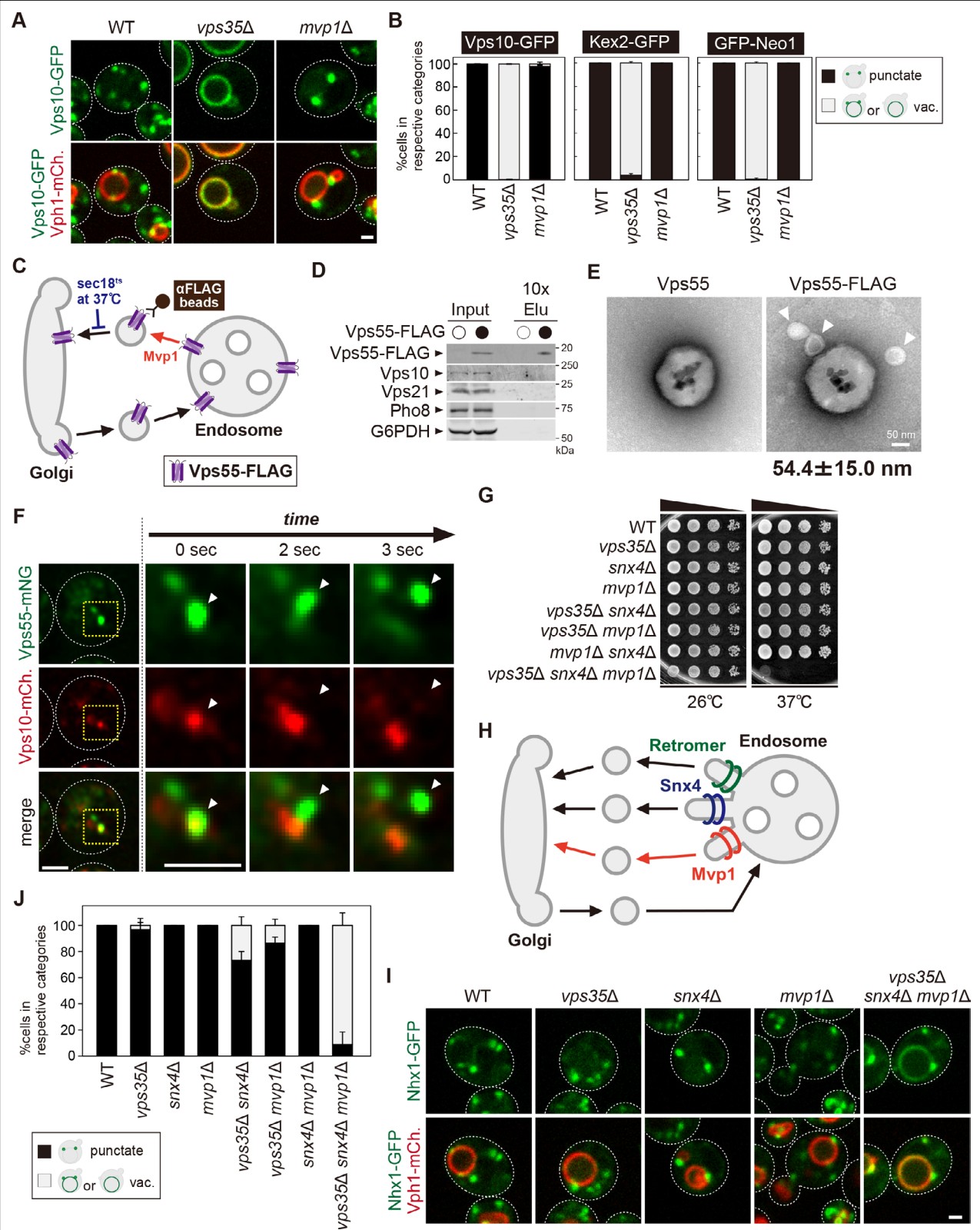

**Figure 5.** Mvp1 mainly mediates retromer-independent endosomal recycling. (**A**) Localization of Vps10-GFP in wild-type (WT), *vps35Δ*, and *mvp1Δ* cells. (**B**) Quantification of Vps10-GFP, Kex2-GFP, and GFP-Neo1 localization from **A** and *Figure 5—figure supplement 1E,F*, respectively. N = >30 cells from three independent experiments. (**C**) Schematic for immunoisolation of Vps55-FLAG-containing structures. (**D**) The immunoisolation of Vps55-containing vesicles. Vps55-FLAG-containing structures were immunoisolated from *sec18*[ts] mutants incubated at 37 °C for 1 hr, and the isolated structures were

*Figure 5 continued on next page*

*Figure 5 continued*

analyzed by immunoblotting using antibodies against FLAG, Vps10 (retromer cargo), Vps21 (endosome), Pho8 (vacuole), and glucose-6-phosphate dehydrogenase (G6PDH) (cytoplasm). (**E**) Electron microscopy (EM) analysis of the isolated Vps55-FLAG-containing structures from **D**. (**F**) Live-cell imaging analysis of Vps55-mNeonGreen and Vps10-mCherry. (**G**) Cell growth in *vps35Δ snx4Δ mvp1Δ* triple mutants. Cells lacking Vps35 as well as Snx4 and Mvp1 were grown at 26°C and 37°C. (**H**) Model of retromer-, Snx4-, and Mvp1-mediated recycling. (**I**) Nhx1 localization in SNX-BAR mutants. (**J**) Quantitation of Nhx1-GFP localization, from **I** and *Figure 5—figure supplement 1M*. N = >30 cells from three independent experiments. Scale bar: 1 μm.

The online version of this article includes the following figure supplement(s) for figure 5:

**Source data 1.** Source data associated with *Figure 5B*.

**Source data 2.** Uncropped gel images of *Figure 5D*.

**Source data 3.** Source data associated with *Figure 5E*.

**Source data 4.** Source data associated with *Figure 5J*.

**Figure supplement 1.** The analysis of the retromer pathway in Mvp1 mutants.

**Figure supplement 1—source data 1.** Uncropped gel images of *Figure 5—figure supplement 1A*.

**Figure supplement 1—source data 2.** Uncropped gel images of *Figure 5—figure supplement 1B*.

**Figure supplement 1—source data 3.** Uncropped gel images of *Figure 5—figure supplement 1J*.

essential for the fusion of recycling vesicles with the Golgi (*Figure 5C*; *Novick et al., 1980*). When we shifted *sec18*[ts] mutants to a non-permissive temperature (37 °C), Vps55-GFP lost its endosomal localization and was distributed in the cytoplasm (*Figure 5—figure supplement 1H*). We incubated Vps55-FLAG-expressing *sec18*[ts] cells at the non-permissive temperature for 60 min and then immunoisolated Vps55-FLAG-containing structures. Vps55-FLAG was concentrated in the isolated product, but Vps10 was not (*Figure 5D*), nor was Vps21 (endosome) or Pho8 (vacuolar membrane). We also analyzed the isolated Vps55-containing structure by electron microscopy (EM). In the EM images, we could observe spherical structures with a diameter of 40–70 nm attached with anti-FLAG magnetic beads (*Figure 5E*). When we eluted these structures from anti-FLAG beads by FLAG peptide, we could observe similar structures by EM (*Figure 5—figure supplement 1I*). Similarly, we immunoisolated Vps10-FLAG-containing vesicles from *sec18*[ts] cells expressing Vps10-FLAG and Vps55-GFP. In the isolated products, Vps10-FLAG was concentrated, but Vps55-GFP was not (*Figure 5—figure supplement 1J*). Next, we performed live-cell imaging of Vps55-mNeonGreen and Vps10-mCherry. The Vps55-mNeonGreen decorated tubules emerged and detached from Vps55-mNeonGreen- and Vps10-mCherry-positive endosomes (*Figure 5F*). On the other hand, we also observed that Vps10-mCherry-positive but Vps55-mNeonGreen-negative tubules also budded from the endosome (*Figure 5—figure supplement 1K*). Based on these results, we concluded that Mvp1 mediates retromer-independent recycling.

The Snx4 complex recycles v-SNAREs from the endosome in a retromer-independent manner. To ask if Mvp1 is involved in Snx4-mediated recycling, we examined the localization of GFP-Snc1. GFP-Snc1 was localized on the PM, Golgi, and endosomes in WT cells, whereas it was sorted into the vacuole lumen in *snx4Δ* cells (*Figure 5—figure supplement 1L*). In *mvp1Δ* cells, it did not show vacuole lumen localization, suggesting Mvp1 is dispensable for Snx4-mediated recycling.

To ask if the retromer, Snx4, and Mvp1 complexes function in parallel pathways, we generated mutants lacking Vps35 as well as Snx4 and Mvp1 and compared the single or combination mutants for growth. Although *vps35Δ*, *snx4Δ*, and *mvp1Δ* single or double mutants alone did not exhibit any observable growth defect at 37 °C, *vps35Δ snx4Δ mvp1Δ* triple mutants failed to grow at 37 °C (*Figure 5G*). These results suggest that retromer, Snx4, and Mvp1 complexes independently function in endosomal recycling (*Figure 5H*). Interestingly, the Na⁺/H⁺ exchanger Nhx1 still localizes on the endosome in *vps35Δ*, *snx4Δ*, and *mvp1Δ* single mutants, but it accumulated on the vacuole membrane in *vps35Δ snx4Δ mvp1Δ* triple mutants (*Figure 5I and J*). In contrast, *vps35Δ snx4Δ* and *vps35Δ mvp1Δ* double mutants only exhibited a partial defect, and a *snx4Δ mvp1Δ* mutant showed no defect. These results suggest that Nhx1 is cooperatively recycled by retromer, Snx4, and Mvp1 complexes, which is consistent with our conclusion that retromer, Snx4, and Mvp1 complexes mediate distinct pathways (*Figure 5H*).

## Retromer, Snx4, and Mvp1 complexes are required for the proper function of the endosome

Retromer-mediated recycling is dispensable for normal growth (*Krsmanovic et al., 2005*). However, *vps35Δ snx4Δ mvp1Δ* triple mutants exhibited a severe growth defect (*Figure 5G*). Hence, we reasoned that general endosomal functions might also be affected in *vps35Δ snx4Δ mvp1Δ* triple mutants. To evaluate this, we examined Mup1 sorting (*Figure 6A*). Mup1 is a methionine permease that localizes to the PM in the absence of methionine (*Lin et al., 2008*), but upon methionine addition, Mup1 is endocytosed, trafficked to endosomes, and sorted into intraluminal vesicles (ILVs) via the ESCRT pathway (*Henne et al., 2011*). Then, the endosome fuses with the vacuole, which delivers Mup1 to the vacuole lumen. To visualize Mup1 sorting, we fused the pH-sensitive GFP variant, pHluorin, to Mup1 (*Miesenböck et al., 1998*). When Mup1-pHluorin is sorted into ILVs or the vacuole lumen, its fluorescence is quenched (*Prosser et al., 2010*). Thus, we can monitor Mup1 sorting by the disappearance of the Mup1-pHluorin signal. After a 60 min treatment with methionine, the fluorescence of Mup1-pHluorin was quenched in the WT cells, whereas Mup1-pHluorin puncta remained stable in *vps35Δ snx4Δ mvp1Δ* cells (*Figure 6B*). We also scored Mup1 sorting by immunoblotting. In WT cells, Mup1-pHluorin was fully processed after 30 min of stimulation, whereas full-length Mup1-pHluorin remained even after 90 min in *vps35Δ snx4Δ mvp1Δ* cells (*Figure 6C*). We examined the sorting of Carboxypeptidases (CPS), which is another transmembrane cargo for the ESCRT pathway (*Odorizzi et al., 1998*). It was partially defective in *vps35Δ snx4Δ mvp1Δ* cells (*Figure 6D*). To ask which SNX-BAR complexes are responsible, we compared Mup1-GFP sorting in single or combination mutants of retromer, Snx4, and Mvp1. Although *vps35Δ*, *snx4Δ*, and *mvp1Δ* single mutants did not show a defect, *vps35Δ snx4Δ mvp1Δ* triple mutants exhibited a strong delay in Mup1 sorting (*Figure 6—figure supplement 1A*). EM analysis revealed that endosomal morphology in *vps35Δ snx4Δ mvp1Δ* cells was altered (*Figure 6E*, *Figure 6—figure supplement 1B*). These observations suggest that retromer, Snx4, and Mvp1 complexes are required for proper function of the endosome.

## Three endosomal recycling pathways cooperatively contribute to maintain appropriate lipid asymmetry

Since the role of endosomal recycling has been characterized by retromer mutants, we hypothesized that analysis of *vps35Δ snx4Δ mvp1Δ* triple mutants might provide new insights into endosomal recycling. For this purpose, we isolated multicopy suppressors of the temperature-sensitive growth phenotype displayed by *vps35Δ snx4Δ mvp1Δ* triple mutants and identified, in addition to Vps35, Snx4, and Mvp1, the P4 type of ATPase Neo1 (*Figure 6F and G*, *Figure 6—figure supplement 1C*). Neo1 can flip phospholipids, especially phosphatidylethanolamine (PE), from the extracellular/lumenal leaflet to the cytoplasmic leaflet of the membrane bilayer, thereby establishing an asymmetric distribution of phospholipids. Neo1 mutants are defective in establishing membrane asymmetry, which leads to hypersensitivity to duramycin, a bioactive peptide that disrupts the membrane through the binding of extracellular PE (*Figure 6H*; *Takar et al., 2016*). To assess the PE asymmetry of the PM, we examined cell growth of *vps35Δ*, *snx4Δ*, and *mvp1Δ* single or combination mutants in the presence of duramycin. Although *vps35Δ*, *snx4Δ*, and *mvp1Δ* single mutants and *snx4Δ mvp1Δ* double mutants only exhibited a mild defect, *vps35Δ snx4Δ* and *vps35Δ mvp1Δ* cells were severely impaired for growth (*Figure 6I*). Also, the *vps35Δ snx4Δ mvp1Δ* triple mutant failed to grow in the presence of duramycin. These results suggest that three endosomal recycling pathways cooperatively contribute to maintain appropriate lipid symmetry.

An appropriate lipid composition/distribution of the PM is essential for cell integrity. Hence, we evaluated the PM integrity of *vps35Δ snx4Δ mvp1Δ* triple mutants under stress conditions. To score cells for loss of integrity, we used propidium iodide, a membrane-impermeable dye. WT cells and *vps35Δ snx4Δ mvp1Δ* triple mutants grown at 26 °C were barely stained (*Figure 6—figure supplement 1D,E*). At 40 °C, only a small population of WT cells was stained, because they were resistant to mild heat stress conditions (2 h at 40 °C). In contrast, most of the *vps35Δ snx4Δ mvp1Δ* triple mutant cells were stained at 40 °C. These results suggest that three endosomal recycling pathways are required to maintain PM integrity under stress conditions.

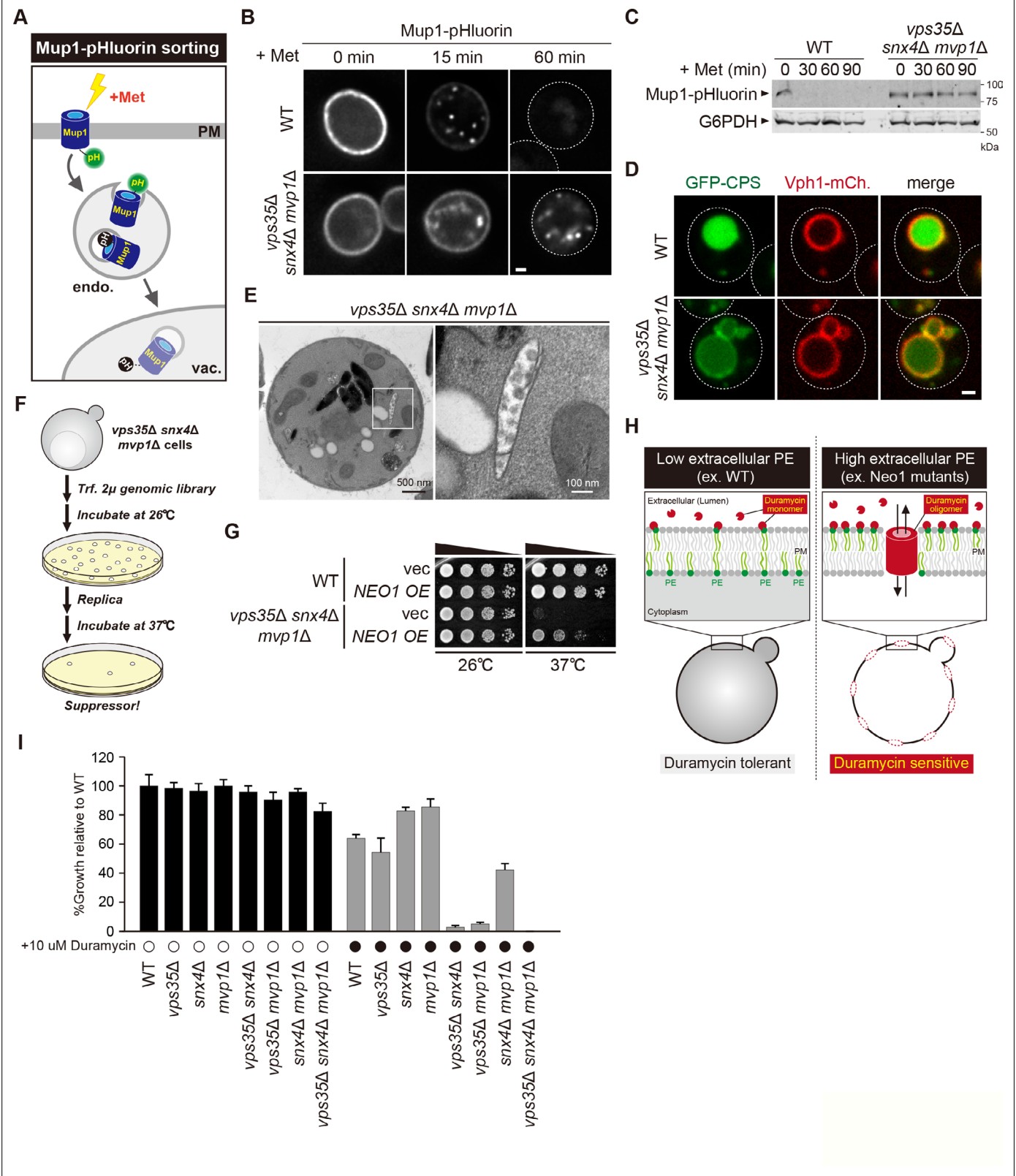

**Figure 6.** Retromer, Snx4, and Mvp1 complexes are required for proper function of the endosome. (**A**) Schematic of Mup1-pHluorin sorting. (**B**) Mup1-pHluorin localization in wild-type (WT) and *vps35Δ snx4Δ mvp1Δ* cells. Cells expressing Mup1-pHluorin were grown to mid-log phase and stimulated with 20 μg/ml methionine. Scale bar: 1 μm. (**C**) Mup1-pHluorin processing in WT and *vps35Δ snx4Δ mvp1Δ* cells. Mup1 sorting was stimulated as in **B**. (**D**) GFP-CPS sorting in WT and *vps35Δ snx4Δ mvp1Δ* cells. (**E**) Thin-section electron microscopy (EM) of an endosome in *vps35Δ snx4Δ mvp1Δ*

*Figure 6 continued on next page*

Figure 6 continued

cells. (**F**) Schematic of screening for multicopy suppressors of *vps35Δ snx4Δ mvp1Δ* triple mutants. (**G**) Growth of *vps35Δ snx4Δ mvp1Δ* triple mutants overexpressing Neo1. (**H**) Schematic of duramycin assay to evaluate extracellular phosphatidylethanolamine (PE). (**I**) Growth of recycling mutants in the presence of duramycin.

The online version of this article includes the following figure supplement(s) for figure 6:

**Source data 1.** Uncropped gel images of *Figure 6C*.

**Source data 2.** Source data associated with *Figure 6I*.

**Figure supplement 1.** The analysis of triple recycling pathway mutants.

**Figure supplement 1—source data 1.** Uncropped gel images of *Figure 6—figure supplement 1A*.

**Figure supplement 1—source data 2.** Source data associated with *Figure 6—figure supplement 1E*.

## Mvp1-mediated endosomal recycling is evolutionarily conserved

The human homolog of Mvp1 is SNX8, which also contains a PX domain and a BAR domain (*Figure 7A*; *Dyve et al., 2009*). SNX8 forms a homodimer and exhibits membrane-deformation activity in vitro (*van Weering et al., 2012*). Although SNX8 has been linked to several diseases, especially AD (*Rosenthal et al., 2012*; *Xie et al., 2019*), its molecular function has not been analyzed in detail. When GFP-SNX8 was expressed in Hela cells, it showed punctate structures that colocalized with the early endosome protein EEA1, as reported previously (*Figure 7B–(i)*; *Dyve et al., 2009*). In addition to these puncta, we also observed tubule-like structures that also were labeled by EEA1 (*Figure 7B -(ii)*). Live-cell imaging analysis revealed that the tubule-like structures budded from the GFP-SNX8-positive endosome (*Figure 7C*, *Figure 7—figure supplement 1A*, *Figure 7—video 1*), which is consistent with a previous report (*van Weering et al., 2012*). SNX1 overexpression induces endosomal swelling and tubulation (*Carlton et al., 2004*). Similarly, when we increased the expression of GFP-SNX8, the endosome marked by GFP-SNX8 was enlarged (*Figure 7—figure supplement 1B, C*). Some of these endosomes have long extended tubule structures (*Figure 7—figure supplement 1B, D, E*). To analyze the biogenesis of this extended tubule structure, we performed live-cell imaging. Once GFP-SNX8 was concentrated on the endosomal surface, the extended tubules emerged from that site (*Figure 7D* and *Figure 7—video 2*). These observations suggest that SNX8 mediates the formation of tubules that bud from the endosome.

## Discussion

This study reveals that the SNX-BAR protein Mvp1/SNX8 assembles with the dynamin-like GTPase Vps1 to mediate endosomal recycling (*Figure 7E*). Mvp1 forms a homodimer and is recruited to the endosome through PI3P binding. The Mvp1 dimer recognizes its cargo, Vps55, through a specific sorting motif. The BAR domain of Mvp1 deforms the endosomal membrane to form cargo-containing tubule-like structures. Subsequently, Mvp1 recruits Vps1 to catalyze membrane scission. Importantly, Vps55 retrieval does not require the retromer or the Snx4 complex. In human cells, SNX8, the human homolog of Mvp1, facilitates the formation of endosomal recycling tubules. Thus, we propose that Mvp1 mediates a conserved endosomal recycling pathway mechanistically distinct from retromer- and Snx4-mediated recycling.

*Chi et al., 2014* proposed that Mvp1 functions in the retromer pathway, but no specific cargo for Mvp1 has been identified. We found that Vps55 (human OB-RGRP) is recycled by Mvp1 in a retromer-independent manner. Consistent with this, Mvp1-dependent sorting of Vps55 requires its 61-YHTSDFM-67 sorting signal, which does not resemble the retromer-binding sequence defined as ØX[L/M/V], where Ø is F/Y/W (*Cullen and Steinberg, 2018*), although it is still possible that the sorting signal also participates in the interaction with Vps68. Through biochemical analysis, we revealed that retromer and Mvp1 mediate the formation of distinct recycling vesicles. Also, although *vps35Δ*, *snx4Δ*, and *mvp1Δ* single mutants did not exhibit any growth defect at 37 °C, *vps35Δ snx4Δ mvp1Δ* triple mutants failed to grow at 37 °C. Based on these results, we propose that Mvp1 mainly functions in retromer-independent recycling, consistent with Mvp1 not being essential for retromer-mediated recycling. However, it does not exclude the possibility that Mvp1 assembles with the retromer. Since Mvp1 and retromer cooperatively recycle Nhx1 and also contribute to the appropriate phospholipid distribution, a subpopulation of Mvp1 molecules may coassemble with the retromer. Indeed, *Chi*

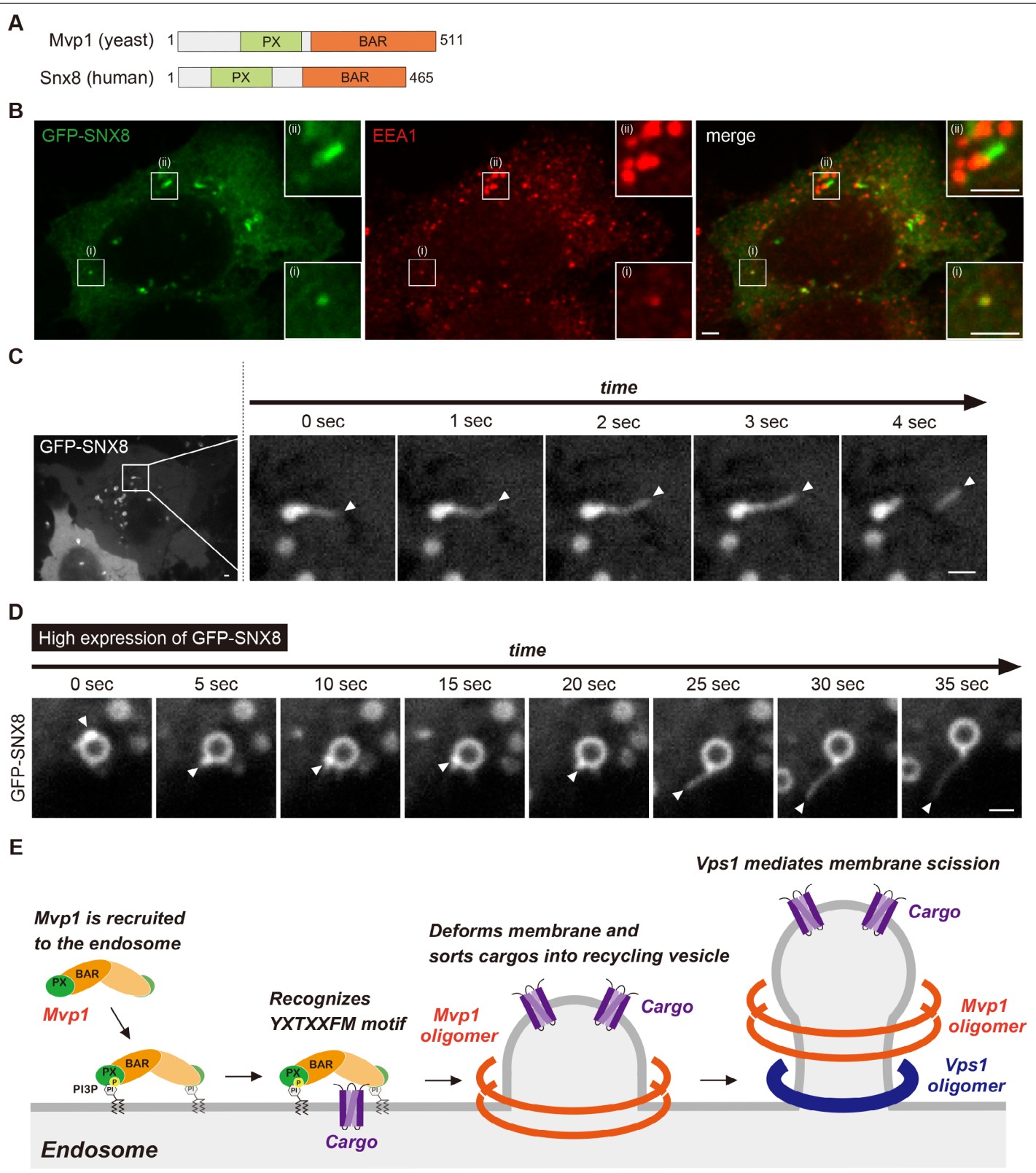

**Figure 7.** Mvp1-mediated endosomal recycling is evolutionarily conserved. (**A**) Schematic of Mvp1 and Snx8. (**B**) Immunofluorescence of GFP-SNX8-expressing Hela cells, with EEA1 serving as an endosomal marker. (**C**) Live-cell imaging of GFP-SNX8. (**D**) Live-cell imaging of highly expressed GFP-SNX8. (**E**) Model of Mvp1-mediated endosomal recycling. Scale bar: 1 μm.

*Figure 7 continued on next page*

*Figure 7 continued*

The online version of this article includes the following video and figure supplement(s) for figure 7:

**Figure supplement 1.** The analysis of GFP-SNX8 tubule structure.

**Figure supplement 1—source data 1.** Source data associated with *Figure 7—figure supplement 1C*.

**Figure supplement 1—source data 2.** Source data associated with *Figure 7—figure supplement 1D*.

**Figure supplement 2.** Highly expressed Vps55 is mislocalized in the retromer mutants.

**Figure 7—video 1.** GFP-SNX8 tubule structure budded from the endosome.

https://elifesciences.org/articles/69883/figures#fig7video1

**Figure 7—video 2.** GFP-SNX8 tubules emerged from their concentration site, related to Figure 7D.

https://elifesciences.org/articles/69883/figures#fig7video2

---

*et al., 2014* reported that 54 % of the endosomal tubule was both Vps17 and Mvp1 positive (33 % Vps17 only, 13 % Mvp1 only). We also observed Vps10- and Vps55-positive tubule structures, but we could not exclude the possibility that these structures resulted in fission of the endosome. To answer this, three-color live-cell imaging of retromer, Mvp1, and the endosomal marker will be required.

*Bean et al., 2017* reported that the appropriate localization of Vps55 also requires the retromer complex, but why was the localization of Vps55 altered in the retromer mutants? Notably, Mvp1 localized on both the endosome and vacuole in *vps35Δ* cells (*Figure 7—figure supplement 2B*), which raised the possibility that the endosome-localized Mvp1 might be reduced in the retromer mutants. Consistent with this idea, when we expressed another copy of Vps55, Vps55-GFP exhibited vacuole localization in *vps35Δ* cells (*Figure 7—figure supplement 2A*). We also examined the PI3P distribution by using GFP fused with the FYVE domain, which specifically binds to the PI3P. It localized on the endosome in the WT cells, whereas mainly on the vacuole membrane in the retromer mutants (*Figure 7—figure supplement 2C*). These observations suggest that PI3P distribution was altered in the retromer mutants, which impairs the localization of Mvp1. Thus, in some conditions, retromer deficiency might indirectly affect Vps55 localization through an inappropriate PI3P distribution.

The abnormal accumulation of Aβ is a hallmark feature of AD. The amyloid precursor protein (APP) is cleaved by β-secretase (BACE) at the endosome, and then its fragments (Aβ) are unconventionally secreted (*Small and Petsko, 2015*). Interestingly, overexpression of SNX8 reduces Aβ levels and rescues cognitive impairment in an APP/PS1 AD mouse model (*Xie et al., 2019*). However, how SNX8 prevents the accumulation of Aβ has not been determined. In this study, we found that Mvp1 and the retromer not only function independently, but also recycle the same cargo (Nhx1). The human retromer complex has been shown to directly or indirectly (possibly through the sortilin receptor) recycle APP and BACE (*Small and Petsko, 2015*). Since defects in this retrieval lead to AD's pathology, SNX8 may also recycle these AD-related membrane proteins.

We propose that Mvp1 recruits Vps1 to mediate membrane scission. Since the endosomal localization of Vps1 was altered in *vps35Δ snx4Δ mvp1Δ* triple mutants, the retromer and Snx4 complexes may also recruit Vps1 to catalyze the fission of recycling vesicles. Future analysis will be necessary to characterize how Vps1 acts on retromer- and Snx4-mediated recycling tubules. In mammalian cells, dynamin mainly localizes not only on the PM, but also to the endosome (*Nicoziani et al., 2000*). *Soreng et al., 2018* reported that SNX18 is responsible for the recruitment of dynamin-2 to the recycling endosome . Inhibiting dynamin causes the tubulation of endosomes (*Mesaki et al., 2011*). Also, the dominant-negative dynamin mutant impairs the endosome-to-Golgi trafficking of Shiga toxin (*Lauvrak et al., 2004*). These observations raise the possibility that dynamin-mediated endosome scission is conserved. However, unlike in yeast cells, the WASH complex, an activator of Arp2/3-dependent actin polymerization, forms a complex with the retromer, which might be sufficient for membrane scission (*Simonetti and Cullen, 2019*). How membrane scission is catalyzed in endosomal recycling pathways is a fundamental question in this field.

Recycling of key membrane proteins (SNAREs, receptors, flippases, transporters, etc) allows cells to maintain organelle identity and function. In this study, we reveal that Mvp1 mediates a novel endosomal recycling pathway. Our findings pave the way toward understanding how cells maintain the composition and function of each organelle. However, several questions remain to be answered. (1) To form a uniform size of recycling vesicles, the activity of Mvp1 and Vps1 should be temporally and spatially regulated, but its regulation mechanism(s) remains unknown. (2) How Mvp1 is uncoated from

the recycling vesicles also requires investigation. (3) How is Vps55 recycled, before endosome fusion with the vacuole, is still unclear. (4) Other membrane proteins and lipids delivered through the Mvp1 pathway need to be identified. (5) How do SNX-BAR family proteins contribute to the appropriate distribution of phospholipids? Future studies should address these critical issues.

## Materials and methods

**Key resources table**

| Reagent type (species) or resource | Designation | Source or reference | Identifiers | Additional information |
|---|---|---|---|---|
| Antibody | Anti-DYKDDDDK (mouse monoclonal) | WAKO | 1E6 | WB (1:2000) |
| Antibody | Anti-GFP (mouse monoclonal) | Roche | Clone 7.1/13.1 | WB (1:5000) |
| Antibody | Anti-GFP (mouse monoclonal) | Santa Cruz | B-2 | WB (1:5000) |
| Antibody | Anti-GFP (rabbit polyclonal) | Torrey Pines Biolabs | | WB (1:5000) |
| Antibody | Anti-HA (mouse monoclonal) | Roche | 12CA5 | WB (1:5000) |
| Antibody | Anti-Myc (mouse monoclonal) | Santa Cruz | 9E10 | WB (1:5000) |
| Antibody | Anti-Vps5 (rabbit polyclonal) | *Horazdovsky et al., 1997* | | WB (1:5000) |
| Antibody | Anti-Vps26 (rabbit polyclonal) | *Reddy and Seaman, 2001* | | WB (1:5000) |
| Antibody | Anti-Vps29 (rabbit polyclonal) | *Seaman et al., 1998* | | WB (1:5000) |
| Antibody | Anti-Vps35 (rabbit polyclonal) | *Seaman et al., 1998* | | WB (1:5000) |
| Antibody | Anti-Vps10 (mouse monoclonal) | Abcam | | WB (1:5000) |
| Antibody | Anti-Vps21 (rabbit polyclonal) | *Horazdovsky et al., 1994* | | WB (1:5000) |
| Antibody | Anti-G6PDH (rabbit polyclonal) | Sigma-Aldrich | | WB (1:20,000) |
| Antibody | Anti-ALP (mouse monoclonal) | NOVEX | | WB (1:1000) |
| Antibody | Anti-Pgk1 (mouse monoclonal) | Invitrogen | | WB (1:10,000) |
| Antibody | Anti-EEA1 (rabbit monoclonal) | Cell Signaling Technology | C45B10 | IF (1:300) |
| Antibody | Anti-Ubiquitin (mouse monoclonal) | Santa Cruz Biotechnology | P4D1 | WB (1:500) |
| Antibody | IRDye 800CW goat anti-mouse | LI-COR | | WB (1:5000) |
| Antibody | IRDye 800CW goat anti-rabbit | LI-COR | | WB (1:5000) |
| Antibody | IRDye 680LT goat anti-rabbit | LI-COR | | WB (1:5000) |
| Antibody | IRDye 680LT goat anti-mouse | LI-COR | | WB (1:5000) |
| Antibody | Goat Alexa Fluor Plus 647 anti-rabbit | Thermo Fisher Scientific | | IF (1:250) |
| Other | N-ethylmaleimide | Acros Organics | 156100050 | |
| Other | Complete Protease Inhibitor Cocktail | Roche | 11697498001 | |
| Other | NHS beads | TAMAGAWA SEIKI | TAS8848 N1141 | |

*Continued on next page*

*Continued*

| Reagent type (species) or resource | Designation | Source or reference | Identifiers | Additional information |
|---|---|---|---|---|
| Other | 3 X FLAG Peptide | Sigma | F4799-25MG | |
| Other | Concanavalin A | Sigma | L7647-250MG | |
| Other | GFP-TRAP_A beads | Chromo Tek | gta-10 | |
| Other | FuGENE HD Transfection Reagent | Promega | E2311 | |
| Other | ProLong Gold Antifade Mountant | Thermo Fisher Scientific | P10144 | |
| Other | PMSF | Sigma | 10837091001 | |
| Other | Saponin | CALBIOCHEM | 558,255 | |
| Other | Triton X-100 | SIGMA | X100-500ML | |
| Other | TALON Metal Affinity Resin | Clontech | 635,502 | |
| Other | SUMO Protease | Milipore | SAE0067-2500UN | |
| Software, algorithm | SoftWoRx | GE Healthcare | | |
| Software, algorithm | SlideBook 6.0 | Intelligent Imaging Innovations | | |
| Software, algorithm | ImageJ | NIH | | |
| Software, algorithm | SnapGene | GSL Biotech | | |

## Yeast strain and media

*Saccharomyces cerevisiae* strains used in this study are listed in *Supplementary file 1A*. Standard protocols were used for yeast manipulation (*Kaiser et al., 1994*). Cells were cultured at 26 °C to mid-log phase in Yeast Extract-Peptone-Dextrose (YPD) medium (1 % (w/v) yeast extract, 2 % (w/v) bacto peptone, and 2 % (w/v) glucose) or Yeast Nitrogen Base (YNB) medium (0.17 % (w/v) yeast nitrogen base w/o amino acids and ammonium sulfate, 0.5 % (w/v) ammonium sulfate, and 2 % (w/v) glucose) supplemented with the appropriate nutrients.

## Mammalian cell line

Hela cells were kindly provided by Dr. Anthony Bretscher (Cornell University). Cell lines were verified to be free of mycoplasma contamination and the identities were authenticated by short tandem repeat (STR) profiling.

## Cell culture conditions for mammalian cells

Hela cells were maintained at 37 °C and 5 % $CO_2$ in Dulbecco's Modified Eagle Medium (DMEM) supplemented with 10 % fetal bovine serum (FBS), penicillin, and streptomycin.

## Plasmids

Plasmids used in this study are listed in *Supplementary file 1B*.

## Fluorescence microscopy

Fluorescence microscopy was performed using a CSU-X spinning-disk confocal microscopy system (Intelligent Imaging Innovations) or a DeltaVision Elite system (GE Healthcare Life system).

A CSU-X spinning-disk confocal microscopy system is equipped with a DMI 6000B microscope (Leica), a ×100/1.45 numerical aperture (NA) objective, and a QuantEM electron-multiplying charge-coupled device (CCD) camera (Photometrics). Imaging for yeast cells was done at room temperature (RT ) in YNB medium using GFP and mCherry channels with different exposure times according to each protein's fluorescence intensity. Imaging for mammalian cells was done at 37 °C in FluoroBride DMEM media (Thermo Fisher) using GFP, mCherry, and far-red-fluorescent dye channels. Images were analyzed and processed with SlideBook 6.0 software (Intelligent Imaging Innovations).

A DeltaVision Elite system is equipped with an Olympus IX-71 inverted microscope, a DV Elite complementary metal-oxide semiconductor camera, a ×100/1.4 NA oil objective, and a DV Light SSI 7 Color illumination system with Live Cell Speed Option with DV Elite filter sets. Imaging was done at RT in YNB medium using GFP and mCherry channels with different exposure times according to each protein's fluorescence intensity. Image acquisition and deconvolution (conservative setting; five cycles) were performed using DeltaVision software softWoRx 6.5.2 (Applied Precision).

## Immunoprecipitation for yeast cells

Anti-FLAG-conjugated magnetic beads were prepared according to the manufacturer's protocol. In brief, N-hydroxysuccinimide ester (NHS) ferrite-glycidyl methacrylate (FG) beads (Tamagawa Seiki) were treated with methanol and then incubated with anti-DYKDDDK antibody (Wako) at 4 °C for 30 min. The magnetic beads were mixed with 1.0 M 2-aminoethanol, pH 8.0, at 4 °C for 16–20 hr to quench the conjugation reaction, washed three times with the bead wash buffer (10 mM 4-(2-h ydroxyethyl)-1-piperazineethanesulfonic acid (HEPES)-NaOH [pH 7.2], 50 mM KCl, 1 mM ethylene-diaminetetraacetic acid (EDTA), and 10 % glycerol), and stored in wash buffer containing 1 mg/ml bovine serum albumin (BSA) (A7030; Sigma-Aldrich).

To examine the Mvp1-Mvp1 interaction, cells expressing Mvp1-FLAG and Mvp1-GFP were grown to mid-log phase and washed twice with wash buffer (50 mM Tris-HCl [pH 8.0], 150 mM NaCl, and 10 % glycerol). The cells were lysed in IP buffer (50 mM Tris-HCl [pH 8.0], 150 mM NaCl, 10 % glycerol, 1 mM phenylmethylsulfonyl fluoride (PMSF), and 1 x protease inhibitor cocktail [Roche]) and lysed by beating with 0.5 mm YZB zirconia beads (Yasui Kikai) for 1 min. IP buffer containing 0.2 % Triton X-100 was added to the lysate (a final concentration of 0.1%), and the samples were rotated at 4 °C for 10 min. The solubilized lysates were cleared at 500x$g$ for 5 min at 4 °C, and the resultant supernatants were subjected to high-speed centrifugation at 17,400x$g$ for 10 min. The cleared supernatants were incubated with pre-equilibrated anti-FLAG-conjugated magnetic beads and rotated at 4 °C for 1 hr. After the beads were washed with wash buffer containing 0.1 % Triton X-100, the bound proteins were eluted by incubating the beads in sodium dodecyl sulphate–polyacrylamide gel electrophoresis (SDS-PAGE) sample buffer at 98 °C for 5 min.

The Mvp1-Vps55 interaction was examined similarly as the Mvp1-Mvp1 interaction with some modifications. Cells expressing Vps55-FLAG and Mvp1-GFP were washed twice with phosphate-buffered saline (PBS) wash buffer (20 mM HEPES-KOH [pH 7.2], 0.2 M sorbitol, 50 mM AcOK, and 2 mM EDTA). Cells were lysed in PBS IP buffer (20 mM HEPES-KOH [pH 7.2], 0.2 M sorbitol, 50 mM AcOK, 2 mM EDTA, and 1 x protease inhibitor cocktail [Roche]) as above. The lysate was solubilized with 1.0 % saponin at 4 °C for 60 min. After centrifugation, the cleared supernatant was incubated with pre-equilibrated anti-FLAG-conjugated magnetic beads and rotated at 4 °C for 4 hr. After the beads were washed with wash buffer containing 1.0 % saponin, the bound proteins were eluted by incubating the beads in elution buffer (0.1 M glycine-HCl [pH 3.0], 1 % Triton X-100) at 4 °C for 30 min. Eluted samples were mixed with 2 x SDS-PAGE sample buffer, then incubated at 42 °C for 5 min.

The Mvp1-Vps1 interaction was examined similarly as the Mvp1-Mvp1 interaction with some modifications. Cells expressing Mvp1-FLAG and Vps1-GFP were washed twice with high-salt wash buffer (50 mM Tris-HCl [pH 8.0], 500 mM NaCl, and 10 % glycerol). Cells were lysed in high-salt IP buffer (50 mM Tris-HCl [pH 8.0], 500 mM NaCl, and 10 % glycerol, 1 x protease inhibitor cocktail [Roche]) as above. The lysate was solubilized with 1.0 % Triton-X 100 at 4 °C for 60 min. After centrifugation, the cleared supernatant was incubated with pre-equilibrated anti-FLAG-conjugated magnetic beads and rotated at 4 °C for 1 hr. After the beads were washed with wash buffer containing 1.0 % Triton X-100, the bound proteins were eluted by incubating the beads in SDS-PAGE sample buffer at 98 °C for 5 min.

To examine the Mvp1-retromer interaction, cells expressing Vps5-FLAG, Vps17-HA, Vps26-Myc, and Mvp1-GFP or cells expressing Mvp1-FLAG and Vps17-HA were washed twice with PBS IP buffer. Cells were lysed in PBS IP buffer as above. The lysate was solubilized with 0.5 % Triton-X 100 for 4 °C for 10 min. After centrifugation, the cleared supernatant was incubated with pre-equilibrated anti-FLAG-conjugated magnetic beads and rotated at 4 °C for 2 hr. After the beads were washed with wash buffer containing 0.5 % Triton X-100, the bound proteins were eluted by incubating the beads in SDS-PAGE sample buffer at 98 °C for 5 min.

### Immunoprecipitation under denature condition for yeast cell lysates

To analyze the ubiquitination status of Vps55, cells expressing Vps55-GFP were washed twice with 400 mM N-ethylmaleimide (NEM). Cells were lysed in urea cracking buffer (50 mM Tris-HCl [pH 8.0], 1 % SDS, 8 M urea, 20 mM NEM, and 1 x protease inhibitor cocktail [Roche]) and lysed by beating with 0.5 mm YZB zirconia beads (Yasui Kikai) for 1 min. High-salt IP buffer with 20 mM NEM and 0.2 % Triton X-100 was added to the lysate, and the samples were rotated at 4 °C for 10 min. The solubilized lysates were cleared at 500x*g* for 5 min at 4 °C, and the resultant supernatants were subjected to high-speed centrifugation at 17,400x*g* for 10 min. The cleared supernatants were incubated with pre-equilibrated GFP-TRAP_A beads (Chromo Tek) and rotated at 4 °C for 1 hr. After the beads were washed with SDS wash buffer (50 mM Tris-HCl [pH 8.0], 250 mM NaCl, 1 % SDS, 4 M urea, and 5 % glycerol), the bound proteins were eluted by incubating the beads in SDS-PAGE sample buffer at 98 °C for 5 min.

### Immunoisolation of Vps55-FLAG- or Vps10-FLAG-containing structures from yeast cells

To immunoisolate Vps55-FLAG- or Vps10-FLAG-containing structures, cells expressing Vps55-FLAG or cells expressing Vps10-FLAG and Vps55-GFP grown at 26 °C were incubated at 37 °C for 60 min before harvest. Cells were washed twice with $H_{25}S_{75}E_5$ buffer (25 mM HEPES-NaOH [pH 7.4], 750 mM sorbitol, and 5 mM EDTA). The cells were lysed in $H_{25}S_{75}E_5$ buffer supplemented with 1 x protease inhibitor cocktail (Roche) and lysed by beating with 0.5 mm YZB zirconia beads (Yasui Kikai) for 1 min. The lysates were cleared at 500x*g* for 5 min at 4 °C twice, and the cleared supernatants were incubated with anti-FLAG-conjugated magnetic beads and rotated at 4 °C for 1 hr. After the beads were washed with $H_{25}S_{75}E_5$ buffer, the immunoisolated structures were eluted by incubating the beads in SDS-PAGE sample buffer at 98 °C for 5 min. For the EM analysis, immunoisolated structures were not eluted, but subjected to negative-staining EM analysis.

### Electron microscopy of immunoisolated Vps55-FLAG-containing structures

Immunoisolated Vps55-FLAG-containing structures on anti-FLAG-conjugated magnetic beads were applied to the carbon-coated electron microscope grid, stained with 2 % ammonium molybdate, and imaged on FEI Morgagni 268 TEM.

### Quantitative analysis of cargo localization

Vps55-GFP/mNeonGreen, Vps10-GFP, Kex2-GFP, and GFP-Neo1 localization was classified into two categories: punctate structures and vacuole membrane localization. Cells having both punctate structures and vacuole membrane localization were classified in the vacuole membrane localization category. For each experiment, at least 30 cells were classified, and the data from three independent experiments were used for the statistical analysis.

### Quantitative analysis of Mvp1-GFP and Vps1-GFP localization

Mvp1-GFP and Vps1-GFP localization was classified into two categories: punctate structures and cytoplasmic localization. For each experiment, at least 30 cells were classified, and the data from three independent experiments were used for the statistical analysis.

### Preparation of yeast cell lysates

Cell lysates were prepared as follows. Cells were grown to mid-log phase at 26 °C. Aliquots of cells were mixed with trichloroacetic acid at a final concentration of 15%, and the mixtures were incubated for 30 min at 4 °C. After centrifugation at 17,400 x*g* for 10 min at 4 °C, the cells were washed once with 100 % acetone and then were lysed in SDS-PAGE sample buffer (60 mM Tris-HCl [pH7.5], 2 % (w/v) SDS, 10 % glycerol, 100 mM dithiothreitol (DTT), and bromophenol blue) by beating with 0.5 mm YZB zirconia beads (Yasui Kikai) for 2 min. The lysates were then heated at 98 °C for 5 min. After centrifugation at 10,000 x*g* for 1 min at RT, supernatants were analyzed by SDS-PAGE and immunoblotting using anti-GFP and anti-G6PDH antibodies.

## Multicopy suppressor screening

The *S. cerevisiae* genomic library used for the suppressor screen was prepared as described previously (*Burda et al., 2002*). The genomic library was transformed in *vps35Δ snx4Δ mvp1Δ* triple mutants. Yeast cells were grown on YPD plates at 26 °C for 2 days. Then, cells were replica plated on YPD plates and incubated at 37 °C for 3 days. Temperature-resistant yeast colonies were selected. Then, plasmids were isolated, amplified, and sequenced.

## Yeast growth assays

Cells grown to mid-log phase at 26 °C were diluted back to $OD_{600} = 0.1$. Then, a 10-fold serial dilution was spotted on growth media and incubated at the indicated temperature for 2–3 days.

## Duramycin assay

Cells grown to mid-log phase at 26 °C were diluted back to $OD_{600} = 0.01$. The cells were grown at 26 °C for 16 hr in the presence or absence of 10 μM duramycin, and $OD_{600}$ of the cells was measured.

## PM integrity assay

Cells grown to mid-log phase at 26 °C were divided in half. One half was incubated at 26 °C for 2 hr, and the other half was shifted to 40 °C for 2 hr. Cells were washed by PBS and were stained with 20 μM propidium iodide for 20 min at RT. Cells were washed twice with PBS and analyzed by a fluorescent microscope.

## Electron microscopy for yeast cells

Cells were grown in YNB medium overnight and 40 ODs of cells were harvested at mid-log growth phase the next morning. To fix, the harvested cells were rinsed in 10 ml sodium cacodylate buffer (0.1 M sodium cacodylate [pH 7.4], 5 mM $CaCl_2$, and 5 mM $MgCl_2$), pelleted, decanted, then resuspended in 5 ml fix buffer (3 % glutaraldehyde, 0.1 M sodium cacodylate [pH 7.4], 5 mM $CaCl_2$, 5 mM $MgCl_2$, and 2.5 % sucrose), and incubated at RT for 1 hr with gentle agitation. After fixing, the cells were washed twice with 5 ml 0.1 M sodium cacodylate (pH 7.4), then rinsed with 5 ml pre-spheroplast buffer (0.1 M Tris-HCl [pH 7.6], 25 mM DTT, 5 mM EDTA, and 1.2 M sorbitol). To prepare for spheroplasting, the cells were re-suspended in 5 ml pre-spheroplast buffer and incubated for 10 min at RT. The cells were then rinsed with 5 ml spheroplast buffer (0.1 M phosphocitrate, 1.0 M sorbitol). The cells were then re-suspended in 1 ml spheroplast buffer containing 0.25 mg/ml zymolyase and incubated at RT for 30 min. After spheroplasting, the cells were gently washed twice with 1 ml staining buffer (0.1 M sodium cacodylate [pH 6.8], 5 mM $CaCl_2$) to remove sorbitol. The cells were embedded in 50 μl of 2 % ultra-low-melt agarose and then cut into ~2 $mm^3$ blocks. The blocks were postfixed/stained in 1 ml osmium staining solution (1 % $OsO_4$, 1 % potassium ferrocyanide, 0.1 M sodium cacodylate [pH 6.8], 5 mM $CaCl_2$, and 10 % formamide) for 1 hr at RT. The blocks were washed four times with water, then stained with 1 ml 1 % uranyl acetate overnight. The blocks were washed four times with water, then dehydrated through a graded series of ethanol: 50%, 75%, 95%, and 2 × 100 % for 10 min each (1 ml). The blocks were transitioned to 1:1 propylene oxide:ethanol for 10 min, then for 2 × 5 min in 100 % propylene oxide (1 ml). The blocks were embedded in 1:1 propylene oxide:epon resin (hard formulation; EMS #14120) and left on a rotator overnight to allow the propylene oxide to evaporate. The blocks were transferred to fresh epon resin and polymerized for 24 hr at 60 °C.

The samples were sectioned at ~70 nM. The sections were poststained with 4 % uranyl acetate for 10 min, then washed in water. Next, the sections were stained in Reynold's lead citrate for 2 min, then washed in water. The sections were imaged using a FEI Morgagni 268 TEM.

## Protein expression and purification

His-SUMO-Vps1-GFP and His-SUMO-Mvp1 were expressed in *Escherichia coli* Rosetta (DE3) by the addition of 250 mM isopropyl β- d-1-thiogalactopyranoside (IPTG) for 16 hr at 16 °C and purified using TALON Metal Affinity Resin (Clontech) according to the manufacturer's protocol. The recombinant proteins were eluted with SUMO protease (Milipore) for Vps1-GFP and Mvp1.

## In vitro binding assay

Purified proteins were incubated with anti-GFP magnetic beads at 4 °C for 30 min. After the beads were washed three times with PBS with 0.1% TX-100, the bound proteins were eluted by incubating the beads in SDS-PAGE sample buffer at 98 °C for 5 min.

## Transfection for mammalian cells

Transient transfections were carried out using FuGENE (Promega) according to the manufacturer's instructions, and the experiments were performed 48 hr after transfection.

## Immunostaining for mammalian cells

Hela cells seeded on coverslips were fixed in 3.7 % formaldehyde/PBS for 10 min at RT, washed with PBS, and permeabilized with 0.2 % Triton-X-100/PBS for 5 min, washed, and blocked with 2 % FBS/PBS for 10 min at RT. Cells were stained for 1 hr at RT with anti-EEA1 antibody in 2 % FBS/PBS. After washing in PBS, cells were stained for 1 hr at RT with Alexa Fluor 647 anti-rabbit antibody in 2 % FBS/PBS. Cells were washed with PBS and mounted using Prolong Gold Hardmount and imaged by a fluorescent microscope.

## Acknowledgements

We appreciate Richa Sardana and Chris Fromme for critical reading of the manuscript. We also thank all Emr lab members for helpful discussions. We deeply appreciate Andrew Lombardo, Risat Zaman, David McDermitt, and Anthony Bretscher for help with establishing the mammalian cell culture experiments. SW Suzuki is supported by JSPS Postdoctoral Fellowships for Research Abroad and the Osamu Hayaishi Memorial Scholarship for Study Abroad. This work was supported by a Cornell University Research Grant (CU563704) to SD Emr.

## Additional information

### Funding

| Funder | Grant reference number | Author |
| --- | --- | --- |
| Cornell University | Research Grant CU3704 | Scott D Emr |

The funders had no role in study design, data collection and interpretation, or the decision to submit the work for publication.

### Author contributions

Sho W Suzuki, Conceptualization, Data curation, Formal analysis, Investigation, Methodology, Project administration, Validation, Visualization, Writing - original draft, Writing - review and editing; Akihiko Oishi, Nadia Nikulin, Jeff R Jorgensen, Investigation; Matthew G Baile, Investigation, Writing - original draft; Scott D Emr, Funding acquisition, Resources, Supervision, Writing - review and editing

### Author ORCIDs

Sho W Suzuki (iD) http://orcid.org/0000-0002-6563-5387
Akihiko Oishi (iD) http://orcid.org/0000-0003-0351-3592
Matthew G Baile (iD) http://orcid.org/0000-0002-2680-1178
Scott D Emr (iD) http://orcid.org/0000-0002-5408-6781

### Decision letter and Author response

Decision letter https://doi.org/10.7554/eLife.69883.sa1
Author response https://doi.org/10.7554/eLife.69883.sa2

## Additional files

### Supplementary files

• Supplementary file 1. Yeast strains and Plasmids.

- Transparent reporting form

## Data availability

All data generated or analyzed during this study are included in the manuscript and supporting files.

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
