## [Decision Letter]

**Acceptance summary:**

This paper will be of broad interest to the vesicle trafficking field, as it implicates an evolutionarily conserved SNX-BAR protein in cargo recognition, carrier formation, and recruitment of a dynamin-like scission factor. An extensive set of experiments reproduces and builds on past work and comes to some new conclusions.

**Decision letter after peer review:**

Thank you for submitting your article "A PX-BAR protein Mvp1/SNX8 and a dynamin-like GTPase Vps1 drive endosomal recycling" for consideration by *eLife*. Your article has been favorably reviewed by 3 peer reviewers, and the evaluation has been overseen by Suzanne Pfeffer as the Senior and Reviewing Editor. The following individuals involved in review of your submission have agreed to reveal their identity: Christian Ungermann (Reviewer #1); Todd R Graham (Reviewer #3).

The reviewers have discussed their reviews with one another, and the Reviewing Editor has drafted this to help you prepare a revised submission. We hope you will consider all the points raised with focus on strengthening the data related to Mvp1 recognition of Vps55 and Vps1 as described below. We hope you will find these comments constructive.

Essential revisions:

1. Show that Mvp1 binds Vps55 cytosolic sequences in vitro, and that this is blocked by the mutation that disrupts sorting, would provide stronger evidence that Mvp1 recognizes and sorts Vps55. Furthermore, transplanting this signal to a heterologous protein would test if this sequence is sufficient for Mvp1-dependent sorting; inclusion of a control cargo that does not bind would also be helpful.

2. Further defining how Mvp1 binds Vps1 would increase the novelty of the study. It is interesting that mutating the Mvp1 dimerization interface didn't block localization to endosomes, but did block sorting. This could mean that Mvp1 oligomerization creates the tubule that recycles Vps55. Alternatively, as suggested by Chi et al., Mvp1 oligomerization could further constrict an existing tubule to promotes the recruitment of Vps1. Can this dimerization mutant recruit Vps1 to endosomes?

3. The authors show that overexpression of Vps55 causes its mislocalization to the vacuole in Figure 3G-J. Does overexpression of Mvp1 or the Vps55-interacting protein Vps68 suppress the overexpressed Vps55 missorting to the vacuole? This would address whether or not one of these proteins is the limiting component for sorting Vps55.

4. Please revise the text throughout to provide a more complete description of the relevant prior literature as described below.

*Reviewer #1 (Recommendations for the authors):*

1. Figure 3,4: The authors nicely dissect the role of Vps1 for the Mvp1 pathway, and Vps1 K42 clearly affects recycling via Mvp1. Is Vps1 required for all three recycling pathways? The authors nicely show that Vps55 is affected in the Vps1 K42A mutant, but should also test the other cargoes (like Atg27).

2. In Figure 6, the authors analyze the triple recycling mutant and its deficiency. Is the same Mup1 trafficking defect and duramycin sensitivity observed in vps1 mutants alone or in combination with snx4 or vps35 deletions? This again would address if Vps1 is required for all three pathways or just for the Mvp1 pathway.

3. Figure 6: The authors show that the triple mutant of mvp1 snx4 and retromer has a strong sensitivity to heat stress (Figure 6). So far, only Vps55 has been characterized as a cargo, whose function remains unclear. Having a recycling deficient mutant of Vps55, the authors could ask if the Vps55 mutant (Figure 3C, most right) combined with snx4 and retromer deletion results in the same phenotype.

4. The authors have evidence that Nhx1 is recognized by all three pathways. It would be nice if the authors could speculate on the sorting motif responsible for general vs. specific (e.g. Mvp1 dependent) recycling.

*Reviewer #2 (Recommendations for the authors):*

Demonstrating that Mvp1 binds Vps55 cytosolic sequences in vitro, and that this is blocked by the mutation that disrupts sorting, would provide stronger evidence that Mvp1 recognizes and sorts Vps55. Furthermore, transplanting this signal to a heterologous protein would test if this sequence is sufficient for Mvp1-dependent sorting.

Further defining how Mvp1 binds Vps1 would increase the novelty of the study. It is interesting that mutating the Mvp1 dimerization interface didn't block localization to endosomes, but did block sorting. This could mean that Mvp1 oligomerization creates the tubule that recycles Vps55. Alternatively, as suggested by Chi et al., Mvp1 oligomerization could further constrict an existing tubule to promotes the recruitment of Vps1. Can this dimerization mutant recruit Vps1 to endosomes?

Figure 3B: Why are there asterisks marking two of the residues? This should be explained in the figure legend. These two residues seem to cause a drastic sorting defect. Why are they not included as important in the model shown in F?

*Reviewer #3 (Recommendations for the authors):*

1. The authors describe Mvp1/SNX8 as a coat protein involved in Vps55 retrieval. One clear role for coat proteins is in the selection of cargo and it would be great to see the data supporting a role for Mvp1 in recognition of cargo sorting signals strengthened. Figure 3E shows a co-IP experiment between Mvp1 and either WT Vps55 or Vps55-4Ala (sorting signal mutant). The difference in the two co-IPs was rather slight and was not quantified. Quantitation of these results and evidence of reproducibility would improve confidence in this result. It would also be worth discussing if the YXTXXFM motif is similar to other sorting signals recognized by retromer or other sorting nexins.

2. Related to the first point – the authors show that overexpression of Vps55 causes its mislocalization to the vacuole in Figure 3G-J. Does overexpression of Mvp1 suppress the overexpressed Vps55 missorting to the vacuole? This would address whether or not Mvp1 is the limiting component for sorting Vps55.

3. Figure 4 shows that Vps1 membrane recruitment is lost in a mvp1 vps35 snx4 triple mutant. This is surprising because I would have assumed Vps1 would also be required for scission of vesicles produced by AP-1, AP-3 and GGA from the Golgi. Perhaps the authors could address this point in the Discussion.

[Editors' note: further revisions were suggested prior to acceptance, as described below.]

Thank you for resubmitting your work entitled "A PX-BAR protein Mvp1/SNX8 and a dynamin-like GTPase Vps1 drive endosomal recycling" for further consideration by *eLife*. Your revised article has been reviewed by 3 peer reviewers, and the evaluation has been overseen by Suzanne Pfeffer as the Senior and Reviewing Editor. For the most part, the reviewers were satisfied with the changes made during revision. However one reviewer felt that additional textual changes would still be required as follows.

(1) The authors were asked to test for direct interaction between Mvp1 and Vps55. Experiments were performed but they failed to detect a direct interaction. They were also unable to provide evidence of sufficiency by transplanting this sequence into a naïve reporter. They have provided better support for the co-IP between these proteins by the new Figure 3—figure supplement 1C. However, they should provide a more cautious interpretation of this result as it is possible that the sorting signal identified in Vps55 is not directly binding to Mvp1. I suggest revising line 156 to include something like "…, although it is also possible that the Vps55-Mvp1 interaction is bridged by another protein." Line 399 in the Discussion should also be revised to "…Mvp1-dependent sorting of Vps55 requires its YHTSDFM sorting signal, which does not…".

The authors' response was disappointing in that the desired outcome was not obtained but the primary conclusions of the study are still valid. I believe the suggested revisions to the text will be sufficient.

(2) I am satisfied with the response to this point regarding the Mvp1-Vps1 interaction.

(3) I do not think the response to this point is adequate. The authors now state that overexpression of Vps68 suppresses the mislocalization phenotype caused by Vps55-GFP overexpression. However, the new data included in the manuscript (Figure 3 – supplement 1E) does not show any difference that I can detect between the vector only and Vps68 OE. No quantitation of this result was provided and it is unclear how the authors came to the conclusion described in the rebuttal. In addition, line 167 of the manuscript states that "Overexpression of Vps68 suppressed degradation of Vps55-GFP (Figure 3 – supplemental 1E),…". There is no evidence of degradation in the data provided. Was this supposed to have been a western blot? If there is actual data to suggest Vps68 is the limiting component, the authors should include in the Discussion the possibility that the Vps55 sorting signal define is actually a Vps68-interaction motif.

(4) The authors have improved their treatment of the literature in the revised manuscript and I think their response to this point is adequate. One more citation they should incorporate into the Discussion is PMID: 29437695, which provides evidence for a sorting nexin recruiting dynamin-2.*Reviewer #2:*

The authors have carried out a considerable number of new experiments in response to the reviewers' comments. In most cases, these experiments have addressed my concerns, and those of the other reviewers. In other instances, I am satisfied that the authors tried hard to carry out the requested experiments even if the results were inconclusive; resolving these issues will require more extensive work that I agree is beyond the scope of this paper. I also appreciate the changes made to the text to recognize others contributions to the field. In conclusion, I am satisfied by this revised version and the authors responses to the previous reviews.

---

## [Author Response]

Essential revisions:1. Show that Mvp1 binds Vps55 cytosolic sequences in vitro, and that this is blocked by the mutation that disrupts sorting, would provide stronger evidence that Mvp1 recognizes and sorts Vps55. Furthermore, transplanting this signal to a heterologous protein would test if this sequence is sufficient for Mvp1-dependent sorting; inclusion of a control cargo that does not bind would also be helpful.

As suggested by the reviewer, we prepared a recombinant cytoplasmic region of Vps55 and Mvp1 from *E. coli* and performed in vitro pull-down assay Author response image 1. However, we were not able to detect the Mvp1-Vps55 interaction (Author response image 1). Because Vps55 is a membrane protein, we also tried to prepare the full length of Vps55 from yeast cells, but we did not obtain enough Vps55 for the in vitro analysis. Characterization of the direct binding between Mvp1 and Vps55 is an essential question. However, to answer this, we might need another expression system (i.e., insect cells, etc.) to prepare a sufficient amount of the full-length Vps55. We feel this is beyond the scope of the current manuscript. We hope that we will be able to answer this in future studies. Since we failed to establish the in vitro binding assay, to address the indirect effect of Vps55 missorting to the vacuole as suggested by reviewer #2, we further examined the Mvp1-Vps55 interaction in vivo by using cells lacking Vam3, which is an essential SNARE for the endosome-vacuole fusion. We confirmed both Vps55-GFP and Vps55^4A^-GFP (recycling sequence mutants) are still localized at the endosome in these mutants (Author response image 1). Strikingly, Mvp1 was coprecipitated from Vps55-FLAG but was not from Vps55^4A^-FLAG (Figure 3—figure supplement 1C was added to the revised manuscript). These results strengthen our conclusion that the Vps55 sorting signal is required for the association with Mvp1.

**Author response image 1. sa2fig1:** 

As suggested by the reviewers, we transplanted the cytoplasmic region of Vps55, including the recycling signal to Vps10 or Ear1, which are endosomal membrane proteins. However, both of them failed to express. It could be because Vps55 has four transmembrane domains, whereas Vps10 and Ear1 only have one. We tried to generate another chimeric protein using tetra-spanning transmembrane proteins, but we could not obtain a suitable TM protein, which is stably localized at the vacuole membrane. Hence, we were not able to address this point.

2. Further defining how Mvp1 binds Vps1 would increase the novelty of the study. It is interesting that mutating the Mvp1 dimerization interface didn't block localization to endosomes, but did block sorting. This could mean that Mvp1 oligomerization creates the tubule that recycles Vps55. Alternatively, as suggested by Chi et al., Mvp1 oligomerization could further constrict an existing tubule to promotes the recruitment of Vps1. Can this dimerization mutant recruit Vps1 to endosomes?

Thank you for your constructive comment. As suggested by the reviewers, we examined that the Vps1-GFP localization in Mvp1 dimerization mutants. In these mutants, Vps1 failed to localize at the endosome (Figure 4—figure supplement 1N was added to the revised version), suggesting that Mvp1 oligomerization is required for the endosomal localization Vps1.

3. The authors show that overexpression of Vps55 causes its mislocalization to the vacuole in Figure 3G-J. Does overexpression of Mvp1 or the Vps55-interacting protein Vps68 suppress the overexpressed Vps55 missorting to the vacuole? This would address whether or not one of these proteins is the limiting component for sorting Vps55.

Thank you for pointing this out. As suggested by the reviewers, we overexpressed Vps68, which suppressed the missorting of the overexpressed Vps55 (Figure 3—figure supplement 1E was added to the revised version). On the other hand, overexpression of Mvp1 did not affect the localization of Vps55 (Author response image 2). These results suggest that Vps68 is the limiting component for sorting Vps55.

4. Please revise the text throughout to provide a more complete description of the relevant prior literature as described below.

Thank you for your constructive comments. We have carefully revised the manuscript (Page 1, line 133; Page 1, line 151; Page 1, line 158; Page 8, line 236; Pge 8, line 244; Page 12, line 377; Page 13, line 399; Page 13, line 408; Page 13, line 414).

Reviewer #1 (Recommendations for the authors):1. Figure 3,4: The authors nicely dissect the role of Vps1 for the Mvp1 pathway, and Vps1 K42 clearly affects recycling via Mvp1. Is Vps1 required for all three recycling pathways? The authors nicely show that Vps55 is affected in the Vps1 K42A mutant, but should also test the other cargoes (like Atg27).

As suggested by the reviewer, we examined Vps10-GFP localization in vps1Δ cells. It was partially missorted to the vacuole membrane (Author response image 3), consistent with a previous study (Chi et al., 2014). We also confirmed the vacuolar localization of Vps10-GFP in *vps1(K42A)* mutants (Author response image 3). To test the Snx4 cargo, we analyzed the localization of GFP-Snc1. It was mislocalized to the vacuole membrane and its lumen (Author response image 3), consistent with the previous publication (Lukehart et al., 2013). These results suggest that all three recycling pathways (Retromer, Snx4, and Mvp1) require Vps1.

**Author response image 3. sa2fig3:** 

2. In Figure 6, the authors analyze the triple recycling mutant and its deficiency. Is the same Mup1 trafficking defect and duramycin sensitivity observed in vps1 mutants alone or in combination with snx4 or vps35 deletions? This again would address if Vps1 is required for all three pathways or just for the Mvp1 pathway.

We examined the Mup1 sorting in *vps1*Δ cells. It was strongly blocked in this mutant (Author response image 4). This result strengthens our interpretation that all three recycling pathways require Vps1, as mentioned above.

**Author response image 4. sa2fig4:** 

3. Figure 6: The authors show that the triple mutant of mvp1 snx4 and retromer has a strong sensitivity to heat stress (Figure 6). So far, only Vps55 has been characterized as a cargo, whose function remains unclear. Having a recycling deficient mutant of Vps55, the authors could ask if the Vps55 mutant (Figure 3C, most right) combined with snx4 and retromer deletion results in the same phenotype.

Since Schluter et al. suggested that Vps55 is required for the MVB sorting (Schluter et al., 2008), we examined Mup1 sorting in *vps55*Δ cells. It was partially defective (Author response image 5). Based on this result and previous publication, we believe that Vps55 is involved in the ESCRT-mediated MVB sorting. How Vps55 functions in the MVB sorting will be the focus of our investigation in the future.

**Author response image 5. sa2fig5:** 

4. The authors have evidence that Nhx1 is recognized by all three pathways. It would be nice if the authors could speculate on the sorting motif responsible for general vs. specific (e.g. Mvp1 dependent) recycling.

Thank you for pointing this out. We did not observe any defect of Nhx1 recycling in each mutant (*vps35*Δ, *snx4*Δ, or *mvp1*Δ), but impaired in *vps35*Δ *snx4*Δ *mvp1*Δ cells. Based on the results, we speculate all three SNX-BAR complexes recognize Nhx1’s sorting signal. Identification of this general sorting signal is an essential question in feature study.

Reviewer #2 (Recommendations for the authors):Demonstrating that Mvp1 binds Vps55 cytosolic sequences in vitro, and that this is blocked by the mutation that disrupts sorting, would provide stronger evidence that Mvp1 recognizes and sorts Vps55. Furthermore, transplanting this signal to a heterologous protein would test if this sequence is sufficient for Mvp1-dependent sorting.

Thank you for pointing this out. As mentioned above, we tried in vitro binding assay and constructed chimeric proteins, but it did not work.

Further defining how Mvp1 binds Vps1 would increase the novelty of the study. It is interesting that mutating the Mvp1 dimerization interface didn't block localization to endosomes, but did block sorting. This could mean that Mvp1 oligomerization creates the tubule that recycles Vps55. Alternatively, as suggested by Chi et al., Mvp1 oligomerization could further constrict an existing tubule to promotes the recruitment of Vps1. Can this dimerization mutant recruit Vps1 to endosomes?

Thank you for pointing this out. As mentioned above, we examined Vps1 localization in Mvp1 dimerization mutants and found that its dimerization is required for Vps1 recruitment (Figure 4—figure supplement 1M was added to the revised manuscript).

Figure 3B: Why are there asterisks marking two of the residues? This should be explained in the figure legend. These two residues seem to cause a drastic sorting defect. Why are they not included as important in the model shown in F?

Sorry for this confusion. These mutants marked with asterisks were more highly expressed than other mutants, which indirectly caused the missorting of Vps55. We carefully revised the Figure legend.

Reviewer #3 (Recommendations for the authors):1. The authors describe Mvp1/SNX8 as a coat protein involved in Vps55 retrieval. One clear role for coat proteins is in the selection of cargo and it would be great to see the data supporting a role for Mvp1 in recognition of cargo sorting signals strengthened. Figure 3E shows a co-IP experiment between Mvp1 and either WT Vps55 or Vps55-4Ala (sorting signal mutant). The difference in the two co-IPs was rather slight and was not quantified. Quantitation of these results and evidence of reproducibility would improve confidence in this result.

As mentioned above, we performed in vitro binding assay, but it did not work.

It would also be worth discussing if the YXTXXFM motif is similar to other sorting signals recognized by retromer or other sorting nexins.

Thank you for pointing this out. We revised the manuscript to reflect this (Page 13, line 399)

2. Related to the first point – the authors show that overexpression of Vps55 causes its mislocalization to the vacuole in Figure 3G-J. Does overexpression of Mvp1 suppress the overexpressed Vps55 missorting to the vacuole? This would address whether or not Mvp1 is the limiting component for sorting Vps55.

Thank you for pointing this out. As suggested by the reviewer, we overexpressed Mvp1, but it did not rescue the Vps55 missorting (Author response image 2). However, Vps68 overexpression did rescue this (Figure 3—figure supplement 1E was added to the revised manuscript), suggesting that Vps68 is a limiting component for Vps55 sorting.

3. Figure 4 shows that Vps1 membrane recruitment is lost in a mvp1 vps35 snx4 triple mutant. This is surprising because I would have assumed Vps1 would also be required for scission of vesicles produced by AP-1, AP-3 and GGA from the Golgi. Perhaps the authors could address this point in the Discussion.

That is an important point. We did not see the colocalization of Vps1-GFP with the TGN localized Sec7-mCherry (Figure 4—figure supplement 1B), suggesting that Vps1 might only be transiently recruited to the TGN. Strikingly, *vps1*Δ cells exhibited the growth defect at 26 degrees (Figure 4—figure supplement 1A), but *vps35*Δ *snx4*Δ *mvp1*Δ cells did not (Figure 5G), suggesting that Vps1 may have another function beyond recycling. This is an important question to be clarified by future studies.

[Editors' note: further revisions were suggested prior to acceptance, as described below.]

For the most part, the reviewers were satisfied with the changes made during revision. However one reviewer felt that additional textual changes would still be required as follows.(1) The authors were asked to test for direct interaction between Mvp1 and Vps55. Experiments were performed but they failed to detect a direct interaction. They were also unable to provide evidence of sufficiency by transplanting this sequence into a naïve reporter. They have provided better support for the co-IP between these proteins by the new Figure —figure supplement 1C. However, they should provide a more cautious interpretation of this result as it is possible that the sorting signal identified in Vps55 is not directly binding to Mvp1. I suggest revising line 156 to include something like "……., although it is also possible that the Vps55-Mvp1 interaction is bridged by another protein." Line 399 in the Discussion should also be revised to "…Mvp1-dependent sorting of Vps55 requires its YHTSDFM sorting signal, which does not…".The authors' response was disappointing in that the desired outcome was not obtained but the primary conclusions of the study are still valid. I believe the suggested revisions to the text will be sufficient.

As suggested by the reviewer, we have carefully revised the manuscript (Line 156; Line 400).

(3) I do not think the response to this point is adequate. The authors now state that overexpression of Vps68 suppresses the mislocalization phenotype caused by Vps55-GFP overexpression. However, the new data included in the manuscript (Figure 3 – supplement 1E) does not show any difference that I can detect between the vector only and Vps68 OE. No quantitation of this result was provided and it is unclear how the authors came to the conclusion described in the rebuttal. In addition, line 167 of the manuscript states that "Overexpression of Vps68 suppressed degradation of Vps55-GFP (Figure 3 – supplemental 1E),…". There is no evidence of degradation in the data provided. Was this supposed to have been a western blot? If there is actual data to suggest Vps68 is the limiting component, the authors should include in the Discussion the possibility that the Vps55 sorting signal define is actually a Vps68-interaction motif.

As suggested by the reviewer, we have added the quantification of “Figure 3—figure supplement 1E” (Figure 3—figure supplement 1F). Also, we have included the possibility suggested by the reviewer in the Discussion (Line 403).

(4) The authors have improved their treatment of the literature in the revised manuscript and I think their response to this point is adequate. One more citation they should incorporate into the Discussion is PMID: 29437695, which provides evidence for a sorting nexin recruiting dynamin-2.

As suggested by the reviewer, we have incorporated suggested paper in the Discussion (Line 442).